



# A Bayesian sequential updating approach to predict phenology of silage maize

Michelle Viswanathan[1], B. Tobias K. D. Weber[1], Sebastian Gayler[1], Juliane Mai[2], Thilo Streck[1]

[1]Institute of Soil Science and Land Evaluation, Biogeophysics, University of Hohenheim, Stuttgart, 70593, Germany
[2]Department of Civil and Environmental Engineering, University of Waterloo, Waterloo, ON N2L 3G1, Canada

*Correspondence to*: michelle.viswanathan@uni-hohenheim.de (Michelle Viswanathan)

**Abstract.** Crop models are tools used for predicting year to year crop development on field to regional scales. However, robust predictions are hampered by factors such as uncertainty in crop model parameters and in the data used for calibration. Bayesian calibration allows for the estimation of model parameters and quantification of uncertainties, with the consideration

of prior information. In this study, we used a Bayesian sequential updating (BSU) approach to progressively incorporate additional data at a yearly time-step to calibrate a phenology model (SPASS) while analysing changes in parameter uncertainty and prediction quality. We used field measurements of silage maize grown between 2010 and 2016 in the regions of Kraichgau and Swabian Alb in southwestern Germany. Parameter uncertainty and model prediction errors were expected to progressively reduce to a final, irreducible value. Parameter uncertainty reduced as expected with the sequential updates.

For two sequences using synthetic data, one in which the model was able to accurately simulate the observations, and the other in which a single cultivar was grown under the same environmental conditions, prediction error mostly reduced. However, in the true sequences that followed the actual chronological order of cultivation by the farmers in the two regions, prediction error increased when the calibration data was not representative of the validation data. This could be explained by differences in ripening group and temperature conditions during vegetative growth. With implications for manual and

automatic data streams and model updating, our study highlights that the success of Bayesian methods for predictions depends on a comprehensive understanding of inherent structure in the observation data and model limitations.

## 1 Introduction

The effects of climate change are already being felt, with increasing global temperature and frequency of extreme events (Porter et al., 2015), which will have an impact on food availability. In order to mitigate risks to food security, suitable

adaptation strategies need to be devised which depend on robust model predictions of the productivity of cropping systems (Asseng et al., 2009). Soil-crop models, which are able to predict changes in crop growth and yield, as a consequence of changes in model inputs like weather, soil properties, and cultivar-specific traits, are considered suitable tools to plan for a secure future. However, achieving robust model predictions is challenging. This is because there is uncertainty in the model inputs, parameters and process representation, as well as in the observations used to calibrate these models (Wallach and

Thorburn, 2017). It is therefore essential to quantify these uncertainties.

Different interpretations of the underlying soil-crop processes have led to different representations in models of varying complexity (Wallach et al., 2016). Process model equations have parameters that represent physiological processes, but are often based on empirical relationships. These relationships describe system processes which cannot be further resolved with reasonable effort. While most parameters, that represent physiological aspects of plant growth and development, can be

determined in dedicated experiments (Craufurd et al., 2013), many others still need to be estimated through model calibration. However, the measured parameters and state variables used for model calibration are uncertain due to errors in the measuring device or technique, as well as the natural variability of the system due to processes occurring at different spatial or temporal scales. Given the different sources of uncertainty, it is important to set up adequate workflows to enable uncertainty quantification and protocols for reporting them, especially when they influence decision-making (Rötter et al.,

40  2011).

For this, the Bayesian approach is an elegant framework to propagate uncertainty from measurements, parameters, and models to prediction. One advantage of Bayesian inference is the use of prior information (Sexton et al., 2016). The posterior probability distribution obtained by conditioning on one dataset can then be used as a prior distribution for the next dataset in a sequential manner (Hue et al., 2008). This approach, called Bayesian sequential updating (BSU), would be more

computationally efficient than having to re-calibrate the model to all previous datasets, every time new data are available. It has been applied to big data studies in which large datasets were split to reduce computational demand and the information was sequentially incorporated (Oravecz et al., 2017). Cao et al., (2016) used BSU to analyse the evolution of the posterior parameter distribution for soil properties by incorporating data from different types of experiments. Thompson et al. (2019) applied this approach to estimate species extinction probabilities where species-siting data were sequential in time. While

there are numerous examples of Bayesian methods being applied in crop modelling for uncertainty quantification and data assimilation (Alderman and Stanfill, 2017; Ceglar et al., 2011; Huang et al., 2017; Iizumi et al., 2009; Makowski, 2017; Makowski et al., 2004; Wallach et al., 2012; Wöhling et al., 2013, 2015), to the best of our knowledge, the BSU method has not yet been evaluated in the field of crop modelling. In this study we assessed whether crop model predictions progressively improve as new information is incorporated using the BSU approach. This ascertains whether the model and parameters are

both temporally and spatially transferable for a particular crop species, an important aspect for large-scale and long-term predictions. Our study was focused on modelling crop phenological development.

Plant phenology is concerned with the timing of plant developmental stages like emergence, growth, flowering, fructification, and senescence. It is controlled by environmental factors such as solar radiation, temperature, water availability, and depends on intrinsic characteristics of the plants (Zhao et al., 2013). Phenological development is a crucial

state variable in soil-crop models, since it controls many other simulated state variables like yield, biomass and leaf area index by influencing the timing of organ appearance and assimilate-partitioning. Phenology is not only species-specific but can also differ between cultivars of the same species (Ingwersen et al., 2018). Model parameters that influence phenology could vary depending on the cultivars (Gao et al., 2020) and possibly also on environmental conditions (Ceglar et al., 2011). Since parameter uncertainty is a major source of prediction uncertainty (Alderman and Stanfill, 2017; Gao et al., 2020), it

impacts prediction quality.

To this end, we assessed the impact of sequentially incorporating new observations with the BSU approach, on prediction quality of phenological development. For this, we modelled phenological development of silage maize grown between 2010 and 2016 in Kraichgau and Swabian Alb, two regions in southwestern Germany with different soil types and climatic conditions. We monitored the changes in parameter uncertainty and evaluated prediction quality by performing model

validation in which simulated phenological development was compared with observations for datasets that were not used for calibration. We hypothesized that:

    (1)   Parameter uncertainty decreases and quality of prediction improves with the sequential updates in which increasing amount of data are used for model calibration.

(2) For the first few sequential updates, the quality of prediction is variable, until the calibration samples become

representative of the population.

(3) The prediction error then progressively drops to an irreducible value that represents the error in inputs,

     measurements, model structure and variability due to spatial heterogeneity that is below model resolution.

We tested these hypotheses by applying BSU in two modelling cases that represent ideal and real-world conditions. In the

first case, we applied BSU to two *synthetic sequences*: an *ideal* sequence of observations wherein the model is able to

simulate the observations accurately, and a *controlled cultivar-environment* sequence of observations which represent

different growing seasons of a single cultivar grown under the same environmental conditions. In the second case, we

applied the BSU to two *true sequences* that follow the actual chronological order in which different cultivars of silage maize

were grown in the two regions under different environmental conditions.

## 2 Materials and Methods

### 2.1 Study sites and measured data

The data used for the study consist of a set of measurements taken at three field sites (site 1, site 2, site 3) in Kraichgau and

two field sites (site 5 and site 6) on the Swabian Alb, in southwestern Germany, between 2010 and 2016 (Fig. 1i) (Weber et

al., 2021). The main crops in rotation were winter wheat, silage maize, winter rapeseed, and cover crops like mustard and

phacelia. Additionally, spelt, spring and winter barley were also grown on the Swabian Alb. Amongst others, continuous

measurements of meteorological conditions, soil temperature and moisture were taken. Soil profiles were sampled at the sites

for characterization of soil properties.

Kraichgau and Swabian Alb represent climatologically contrasting regions in Germany. Kraichgau is situated 100 to 400m

above sea-level and characterized by a mild climate with a mean temperature above 9°C and mean annual precipitation of

720 to 830mm. It is one of the warmest regions in Germany. The Swabian Alb is located at 700 to 1000m above sea-level

with a mean temperature of 6 to 7°C and mean annual precipitation of 800 to 1000mm. Kraichgau soils have often developed

from several metres of Holocene loess, underlain by limestones. They are predominantly Regosols and Luvisols. The

Swabian Alb has a karst landscape with clayey loam soils, often classified as Leptosols. Soils may be less than 0.3m thick in





some areas. While the soils at the sites in Kraichgau are alike, they vary across the sites on the Swabian Alb (Wizemann et al., 2015).


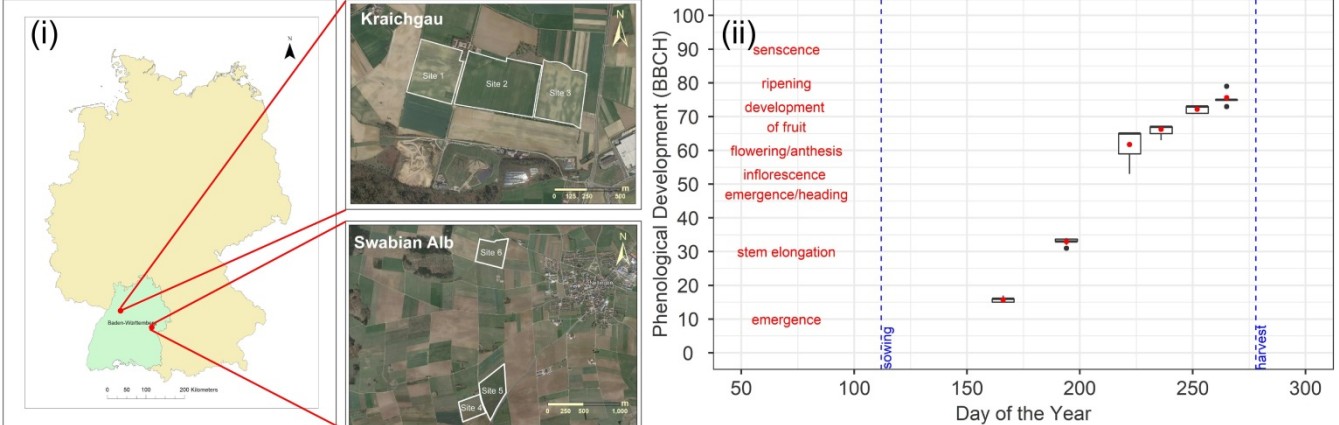

**Figure 1: i) Location of the sites in Kraichgau (site 1, site 2 and site 3) and the Swabian Alb (site 5 and site 6) in the state of Baden-Wuerttemberg, Germany (© Google Earth 2018 modified from Eshonkulov et al., 2019) ii) Observations of phenological development (expressed in BBCH growth stages) of silage maize at site 6 are plotted against the day of the year in 2010. The red labels indicate important phenological development stages. The red points are means of the observations while the box and whiskers represent the range of replicate observations. Length of the box represents the inter-quartile range (IQR), whiskers extend from the box up to 1.5 × IQR and values beyond this range are plotted as points.**

At every study site, which had an area of around 15 ha, replicate observations were made by assessing phenological development stages from maize plants in five subplots of 2m × 2m each. Ten maize plants were chosen from each subplot.

We used the BBCH growth stage code (Meier, 1997) to define the development stages. The BBCH value of 10 marks emergence and the start of leaf development, 30 stands for stem elongation, 50 for inflorescence, emergence or heading, 60 for flowering or anthesis, 70 for development of fruit, 80 for ripening and 90 for senescence (Fig. 1ii). In the following sections, the individual growing seasons for silage maize are denoted by the site and year of growth i.e. the site-year (Table 1). For example, silage maize grown at site 2 in Kraichgau in the year 2012 is referred to as '2_2012'. The different cultivars

used in the study can be grouped into three maturity groups, based on their timing of ripening. Mid-early (ME) and late (L) ripening cultivars were grown in Kraichgau, and early (E) and mid-early (ME) ripening cultivars were grown on the Swabian Alb.





**Table 1: Early (E), mid-early (ME), and late (L) ripening cultivars of silage maize, with their sowing and harvest dates, grown at the study sites in Kraichgau (sites 1, 2 and 3) and the Swabian Alb (sites 5 and 6) between 2010 and 2016.**

| Region | Year | Site | Site-year | Cultivar | Maturity/ Ripening group | Sowing date (DD/MM/YYYY) | Harvest date (DD/MM/YYYY) |
|--------|------|------|-----------|----------|--------------------------|--------------------------|---------------------------|
| Kraichgau | 2011 | 3 | 3_2011 | Canavaro | L | 18/04/2011 | 03/10/2011 |
| Kraichgau | 2012 | 2 | 2_2012 | Canavaro | L | 02/05/2012 | 19/09/2012 |
| Kraichgau | 2014 | 1 | 1_2014 | Grosso | ME | 12/04/2014 | 09/10/2014 |
| Kraichgau | 2014 | 2 | 2_2014 | Grosso | ME | 11/04/2014 | 08/10/2014 |
| Swabian | 2010 | 6 | 6_2010 | Fernandez PR 39 A 98 | ME | 23/04/2010 | 06/10/2010 |
| Swabian | 2011 | 5 | 5_2011 | Agro-Yoko | ME | 25/04/2011 | 04/10/2011 |
| Swabian | 2012 | 5 | 5_2012 | Amanatidis | E | 28/04/2012 | 07/10/2012 |
| Swabian | 2013 | 6 | 6_2013 | SY Kairo & Agro Yoko | ME | 26/04/2013 | 04/10/2013 |
| Swabian | 2015 | 5 | 5_2015 | LG 30.217 | E | 22/04/2015 | 14/09/2015 |
| Swabian | 2016 | 5 | 5_2016 | LG 30.217 | E | 07/05/2016 | 27/09/2016 |
| Swabian | 2016 | 6 | 6_2016 | Toninio | ME | 03/05/2016 | 23/09/2016 |


## 2.2 Soil-crop model

To simulate the soil-crop system, we used the SPASS crop growth model (Wang, 1997). SPASS is implemented in the

Expert-N 5.0 (XN5) software package (Heinlein et al., 2017; Klein et al., 2017; Priesack, 2006). In XN5, the SPASS crop

model is coupled to the Richards equation for soil-water movement as implemented in the model Hydrus-1D (Šimůnek et al.,

1998). The routine uses van Genuchten-Mualem hydraulic functions (van Genuchten, 1980; Mualem, 1976) and the heat

transfer scheme from the Daisy model (Hansen et al., 1990). In the SPASS model, germination to emergence (up to BBCH

10), the vegetative phase (between BBCH 10 to 60) and generative phase (BBCH 61 onwards) of the crop are modelled.

Temperature and photoperiod are the two main factors affecting phenological development rate (for details refer to Appendix

A: SPASS phenology model).



Daily weather data consisting of daily global radiation, maximum and minimum temperatures were used in XN5 to calculate

the air temperatures within the crop canopy. Soil properties (texture class, grain size, rock fraction, bulk density, porosity), as

well as van Genuchten parameters and hydraulic properties (soil water content at wilting point, field capacity, residual and

saturated water content, and saturated hydraulic conductivity) were based on soil samples taken at the sites in 2008 to

characterise the soil profile. The soil horizons in the model were based on these soil profile descriptions. Initial values of soil

volumetric water content were based on measurements. The simulations for each site-year were started on the harvest date of

the preceding crop in the crop rotation at that site. This ensured adequate spin-up time prior to the simulation of silage maize,

which was sown in between April and May.

### 2.3 Selection of model parameters

Parameters were first pre-selected (Hue et al., 2008; Makowski et al., 2006) based on expert knowledge. Parameter values

were elicited from an expert and are given in Table 2. A global sensitivity analysis using the Morris method (Morris, 1991)

was then carried out to identify the sensitive parameters to be estimated through Bayesian calibration (supplementary

material 1). The sensitive parameters identified for calibration were: effective sowing depth (SOWDEPTH), which

influences the emergence rate, and parameters affecting development in the vegetative phase (PDD1, TMINDEV1,

DELTOPT1, and DELTMAX1). Parameter DELTOPT2, from the temperature response function during the reproductive

phase, was estimated during calibration even though it was less sensitive. The choice of using this parameter during

calibration was based on knowledge of model behaviour, so as to reduce the calibration error in the reproductive phase

(Lamboni et al., 2009). Thus, out of eleven pre-selected parameters (Table 2) six were estimated in BSU, while the

remaining parameters were fixed at their expected values.

**Table 2: SPASS model parameters for phenological development. The expected value and two standard deviations (+/-2sd) were**
**elicited from the expert. Column 'Status in calibration' indicates the parameters which were estimated or fixed to the expected**
**value during Bayesian calibration. Minimum (min) and maximum values (max) were set for estimated parameters to constrain the**
**prior parameter ranges to reasonable values during calibration.**

| Parameter name | Description | Unit | Expected value | -2sd | +2sd | min | max | Status in calibration |
|---|---|---|---|---|---|---|---|---|
| PDD1 | Physiological development days from emergence to anthesis | d | 45 | 32 | 60 | | | estimated |
| PDD2 | Physiological | d | 36 | 25 | 60 | | | fixed |

 

| | | | | | | | | |
|---|---|---|---|---|---|---|---|---|
| | development days from anthesis to maturity | | | | | | | |
| PDL | Photoperiod sensitivity factor | - | 0 | 0 | 0.1 | | | fixed |
| DLOPT | Optimal photoperiod length | h | 12 | 10 | 15 | | | fixed |
| TMINDEV1 | Minimum temperature of vegetative development | °C | 6 | 5 | 8 | 0 | 10 | estimated |
| DELTOPT1 | Difference between optimum and minimum temperatures of vegetative development | °C | 28 | 22 | 31 | 1 | 35 | estimated |
| DELTMAX1 | Difference between maximum and optimum temperatures of vegetative development | °C | 10 | 4 | 14 | 1 | 16 | estimated |
| TMINDEV2 | Minimum temperature of reproductive development | °C | 8 | 6 | 10 | | | fixed |
| DELTOPT2 | Difference between optimum and minimum temperatures of reproductive development | °C | 26 | 17 | 32 | 1 | 35 | estimated |
| DELTMAX2 | Difference between maximum and optimum temperatures of reproductive development | °C | 10 | 4 | 14 | | | fixed |
| SOWDEPTH | Effective sowing depth of the seeds in the soil | cm | 8 | 5 | 15 | 1 | 20 | estimated |

## 2.4 Bayesian sequential updating

In the Bayesian sequential updating (BSU) approach, Bayesian calibration is applied in a sequential manner. New data are
used to re-calibrate the model, conditional on the prior information from previously gathered data. We describe the details of this approach below.





Bayes theorem states that the posterior probability of parameters θ given the data Y, P(θ|Y), is proportional to the product of

the joint prior probability of the parameters P(θ) and the probability of generating the observed data with the model, given

the parameters P(Y|θ). The term P(Y|θ) is referred to as the likelihood function and is defined as the likelihood that

observation Y is generated by the model using the parameter vector θ. The posterior probability distribution is obtained by

normalizing this product by the prior predictive distribution (Gelman et al., 2014) or Bayesian Model Evidence (Schöniger et

al., 2015) P(Y), which is obtained by integrating the product over the entire parameter space.

Hence, we write:

$$P(\theta|Y) = \frac{P(\theta)P(Y|\theta)}{P(Y)} \tag{1}$$

where

$$P(Y) = \int_\theta P(\theta)P(Y|\theta)d\theta \tag{2}$$

Equation (2) can become intractable, especially with a large number of parameters as this involves integrating over high

dimensional space (Schöniger et al., 2015). Instead, sampling methods like Markov Chain Monte Carlo (MCMC) are used to

estimate the posterior distribution.

For one site-year $sy_1$ and corresponding observation vector $Y_{sy_1}$, the posterior parameter probability distribution is:

$$P\left(\theta|Y_{sy_1}\right) = \frac{P(\theta)\,P(Y_{sy_1}|\theta)}{\int_\theta P(\theta)\,P(Y_{sy_1}|\theta)\,d\theta} \tag{3}$$

where P(θ) represents the initial prior probability distribution that could be based on expert knowledge. The posterior

parameter distribution $P\left(\theta|Y_{sy_1}\right)$ can now be used as a prior distribution for the next site-year $sy_2$. Thus, for site-year $sy_n$

with an observation vector $Y_{sy_n}$, the posterior parameter probability distribution is:

$$P\left(\theta|Y_{sy_n}\right) = \frac{P\left(\theta|Y_{sy_{(n-1)}}\right) P(Y_{sy_n}|\theta)}{\int_\theta P\left(\theta|Y_{sy_{(n-1)}}\right) P(Y_{sy_n}|\theta)\,d\theta} \tag{4}$$

This equation defines the Bayesian sequential updating (BSU) approach in which the model is calibrated in a sequential

manner. New data from a site-year $(Y_{sy_n})$ is used to re-calibrate the model, conditional on the prior information from





previous site-years. The posterior distribution obtained from the previous Bayesian calibration $P(\theta|Y_{sy_{(n-1)}})$ is used as prior

probability for calibration to the next site-year.

With the aim of making the computations tractable, we deviate slightly from this pure BSU approach as we do not strictly

use the posterior from the previous site-year as the prior for the next, but sequentially calibrate the model to data from

increasing number of site-years instead. The reason for this deviation is that, in applying BSU, where the posterior parameter

distribution is estimated by sampling methods, a probability density function needs to be approximated from the sample, so

that it can be used as a prior probability for the subsequent site-year. This approximation introduces additional errors. Since

joint inference is known to be better than sequential inference using posterior approximations (Thijssen and Wessels, 2020),

Eq. (4) can be re-written, under the assumption that the observations from all site-years are independent and identically

distributed (Gelman et al., 2014), as follows:

$$P(\theta|Y_{sy_n}) = \frac{P(\theta)\ \prod_{x=sy_1}^{sy_n} P(Y_x|\theta)}{\int_\theta\ P(\theta)\ \prod_{x=sy_1}^{sy_n} P(Y_x|\theta)\ d\theta} \tag{5}$$

Thus, we use Eq. (5) to sequentially update the probability distribution of parameters by increasing the dataset size at each

step through the addition of one site-year worth of new data $Y_x$ to the previous dataset $Y_{x-1}$.

After each inferential step, the phenological development at the next site-year $sy_{n+1}$ is predicted by:

$$P(Y_{sy_{n+1}}|Y_{sy_n}) = \int P(Y_{sy_{n+1}}|\theta)P(\theta|Y_{sy_n})\ d\theta \tag{6}$$

where $P(Y_{sy_{n+1}}|Y_{sy_n})$ is the posterior predictive distribution (Gelman et al., 2014). All calculations and the BSU was carried

out using the R programming language (R Core Team, 2020).

In the following sections, we describe the components of Bayes formula in detail.

**2.4.1 Likelihood function**

Let $\theta = (\varphi_1, \varphi_2, \varphi_3, \dots \varphi_j)$ represent a vector of the model parameters to be estimated in this study (Table 2). Suppose

$Y = (\bar{y}_1, \bar{y}_2, \bar{y}_3, \dots \bar{y}_d)$ is a vector of the means of observed phenological development at different days during the growing

season for a particular site-year. The mean observation $\bar{y}_d$ on day d for the site-year is given by:



$$\bar{y}_d = \frac{1}{P}\frac{1}{R}\sum_{p=1}^{P}\sum_{r=1}^{R} y_{r,p,d} \tag{7}$$

where $y_{r,p,d}$ represents the $r^{th}$ replicate of observed phenological development, measured at subplot p on day d for a particular site-year, R is the total number of replicates at subplot p, and P is the total number of subplots per field.

If we assume that all replicates R in all subplots P are independent, the standard deviation of the replicate observations on day d is $\sigma_{r,p,d} = \sqrt{\sum_{p=1}^{P}\sum_{r=1}^{R}\left(y_{r,p,d} - \bar{y}_d\right)^2 / (P \times R)}$ . This is one source of observation error that represents the spatial variability at the study site which is below the spatial resolution of the model. We also assume an additional source of error in identification of the correct phenological stage and its exact timing of occurrence. We assume that this error is within a standard deviation of 2 BBCH ($\sigma_{ident,d} = 2$ for each observation day d). This assumption was made, since 2 is the most common difference between development stages in the phenological development of maize on the BBCH scale. Assuming that the error from replicate observations ($\sigma_{r,p,d}$) and error in the identification of phenological stages are additive, the total observation error is $\sigma_d^2 = (\sigma_{r,p,d} + \sigma_{ident,d})^2$.

The model residual $\bar{y}_d - f(\theta)_d$ is the difference between the observed $\bar{y}_d$ and the model simulated $f(\theta)_d$ phenological stage and is represented by the likelihood function. Assuming normally distributed residuals, it is given by:

$$P(\bar{y}_d|\theta) = \frac{1}{\sigma_d\sqrt{2\pi}}\, e^{-0.5\left(\frac{\bar{y}_d - f(\theta)_d}{\sigma_d}\right)^2} \tag{8}$$

The likelihood values for all the observations are combined by taking the product of the likelihoods per day of observation, under the assumption of independent and identically distributed model residuals. The combined likelihood function is given by:

$$P(Y_x|\theta) = \prod_{d=1}^{D} P(\bar{y}_d|\theta) \tag{9}$$

where $Y_x$ is the observation vector for site-year x.





### 2.4.2 Prior probability distribution

As prior information, we used a weakly informative probability distribution function (pdf) to ensure that the posterior

parameter distributions are mainly determined by the data that are sequentially incorporated. For this, we used a platykurtic

prior probability distribution that is a convolution of a uniform and a normal distribution of the form:

$$
P(\varphi_j) = \begin{cases} \dfrac{1}{c}\dfrac{1}{\sigma\sqrt{2\pi}}e^{-\frac{(x-\mu)^2}{2\sigma^2}} & \text{for } a \le x < \mu - 2\sigma \\[2mm] \dfrac{1}{c}\dfrac{1}{\sigma\sqrt{2\pi}}e^{-2} & \text{for } \mu - 2\sigma \le x \le \mu + 2\sigma \\[2mm] \dfrac{1}{c}\dfrac{1}{\sigma\sqrt{2\pi}}e^{-\frac{(x-\mu)^2}{2\sigma^2}} & \text{for } \mu + 2\sigma < x \le b \end{cases} \tag{10}
$$

where x is a value of the $j^{th}$ model parameter $\varphi_j$ in the parameter vector θ, a and b are the minimum and maximum limit for

the parameter, respectively, μ is the mean (expected value in Table 2), and σ the standard deviation. The normalization

constant c is used to ensure that the area under the curve is unity as required for probability density functions.

$$
c = -\,\text{erf}(\sqrt{2}) + \frac{4}{\sqrt{2\pi}}e^{-2} - \frac{1}{2}\text{erf}(\frac{(a-\mu)}{\sigma\sqrt{2}}) + \frac{1}{2}\text{erf}(\frac{(b-\mu)}{\sigma\sqrt{2}}) \tag{11}
$$

The joint prior pdf was calculated by $P(\theta) = \prod_{j=1}^{J} P(\varphi_j)$ and the model parameters were assumed to be uncorrelated, i.e. the

off-diagonal elements of the variance-covariance matrix are zero. The parameters a, b, c, σ, μ, of $P(\varphi_j)$ was based on expert

knowledge (Table 2).

### 2.4.3 Posterior probability distribution

The posterior parameter distribution was sampled using the Markov Chain Monte Carlo method – Metropolis algorithm

(Metropolis et al., 1953) (for details refer to Appendix B: Posterior sampling using MCMC Metropolis algorithm). Three

chains were run in parallel. A symmetrical transition kernel was chosen as the jump distribution. The jump size was adapted

so that the acceptance rate would be between 25% and 35% (Gelman et al., 1996; Tautenhahn et al., 2012). Convergence of

the chains after jump adaptation was checked using the Gelman-Rubin convergence criteria (<= 1.1) (Brooks and Gelman,

1998; Gelman and Rubin, 1992).





For model validation, the posterior predictive distribution was used to simulate phenological development and compare with observations at site-years that were not included in the calibration sequence.

**2.5 Performance metrics**

230 Bias and normalized root mean square error (NRMSE), as defined in Eq. (12) and (13), for site-year sy were calculated to assess the calibration and prediction performance.

$$\text{Bias}_{sy} = \frac{1}{D} \sum_{d=1}^{D} (\overline{y}_d - f(\theta_i)_d) \tag{12}$$

$$\text{NRMSE}_{sy} = \sqrt{\frac{1}{D} \sum_{d=1}^{D} \frac{(\overline{y}_d - f(\theta_i)_d)^2}{\sigma_d^2}} \tag{13}$$

Here, $\theta_i$ is the i[th] parameter vector, D is the total number of observation days for the particular site-year, $f(\theta_i)_d$ is the simulated phenological development, $\overline{y}_d$ is the mean observed phenological development and $\sigma_d$ is the standard deviation of the observations (as defined in section 2.4.1 Likelihood function) on day d. Under the assumption of normally distributed

235 error, the natural logarithm of the likelihood probability is proportional to the normalized mean square error: $\ln\left(P(Y_{sy}|\theta_i)\right) \propto \text{NRMSE}_{sy}^2$. The normalized bias $\text{NBias}_{sy} = \frac{1}{D}\sum_{d=1}^{D} \frac{\overline{y}_d - f(\theta_i)_d}{\sigma_d}$ is also reported in some plots.

The prediction quality is good when NRMSE is low and bias is zero. Prediction performance is classified as good, moderate, or poor depending on the median NRMSE of the predictions for a site-year. We use the following categories: good performance for median NRMSE <=1, moderate for 1< median NRMSE<=2, poor for 2<median NRMSE<=3 and very poor

240 for median NRMSE>3.

We estimated the information entropy of the posterior parameter distributions after each sequential update using the redistribution estimate equation (Beirlant et al., 1997) (supplementary material 2). A change in entropy with sequential updates indicates a change in uncertainty of the parameters, where higher information entropy indicates greater uncertainty in the posterior parameters. In line with our hypotheses, we expect the entropy to decrease with sequential updates.





### 2.6 Modelling cases

The BSU approach described above and the subsequent analysis using the performance metrics were applied to two *synthetic sequences* and two *true sequences* of site-years. The synthetic sequences were used to demonstrate the application of the BSU approach in ideal conditions, while the true sequences were used to extend the application to real-world conditions. Figure 2 shows the four sequences and the site-years used for calibration and validation.

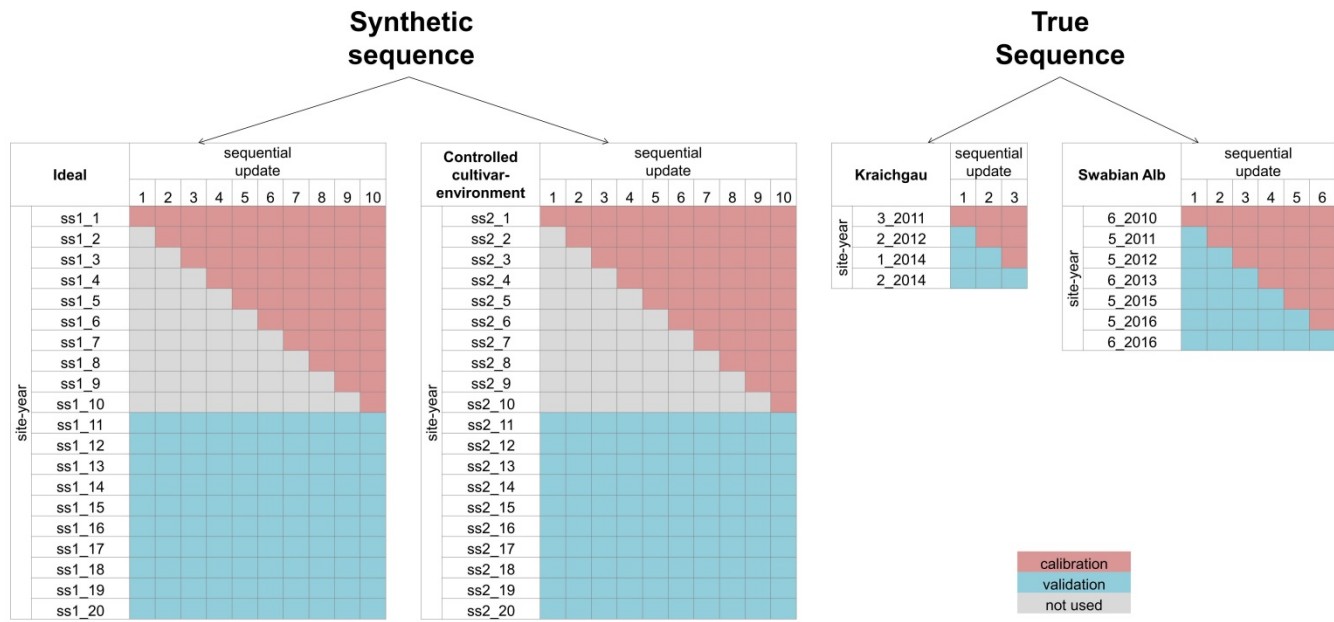

**Figure 2: The site-years used for calibration and validation in each sequential update for the two synthetic sequences namely, ideal and controlled cultivar-environment, and the two true sequences for Kraichgau and the Swabian Alb are shown. In the synthetic sequences, a total of 10 updates were done by sequentially adding 1 through 10 site-years to the calibration dataset. After each update, prediction quality was analysed for a set of 10 validation site-years. A total of 3 sequential updates in Kraichgau and 6 sequential updates in the Swabian Alb true sequences were analysed. In the sequential updates for the true sequences, a site-year was included for calibration, following the actual chronological order of growth. The remaining site-years grown in the region were then used for validation.**

### 2.6.1 Synthetic sequences

We set up two synthetic sequences, namely *ideal* and *controlled cultivar-environment*. In each synthetic sequence, we used 10 sequential updates wherein one through 10 site-years were used in calibration. After each sequential update, the calibrated model was validated against a different set of 10 synthetic site-years (Fig. 2). Note here that the 10 site-years used for validation were the same across the sequential updates. Data from the 10 site-years used for calibration (red box-plots) and





the 10 site-years used for validation (blue box-plots) for the two synthetic sequences are shown in Fig. 3. Site-year 6_2010 was used to generate data for the synthetic sequences, as described below.

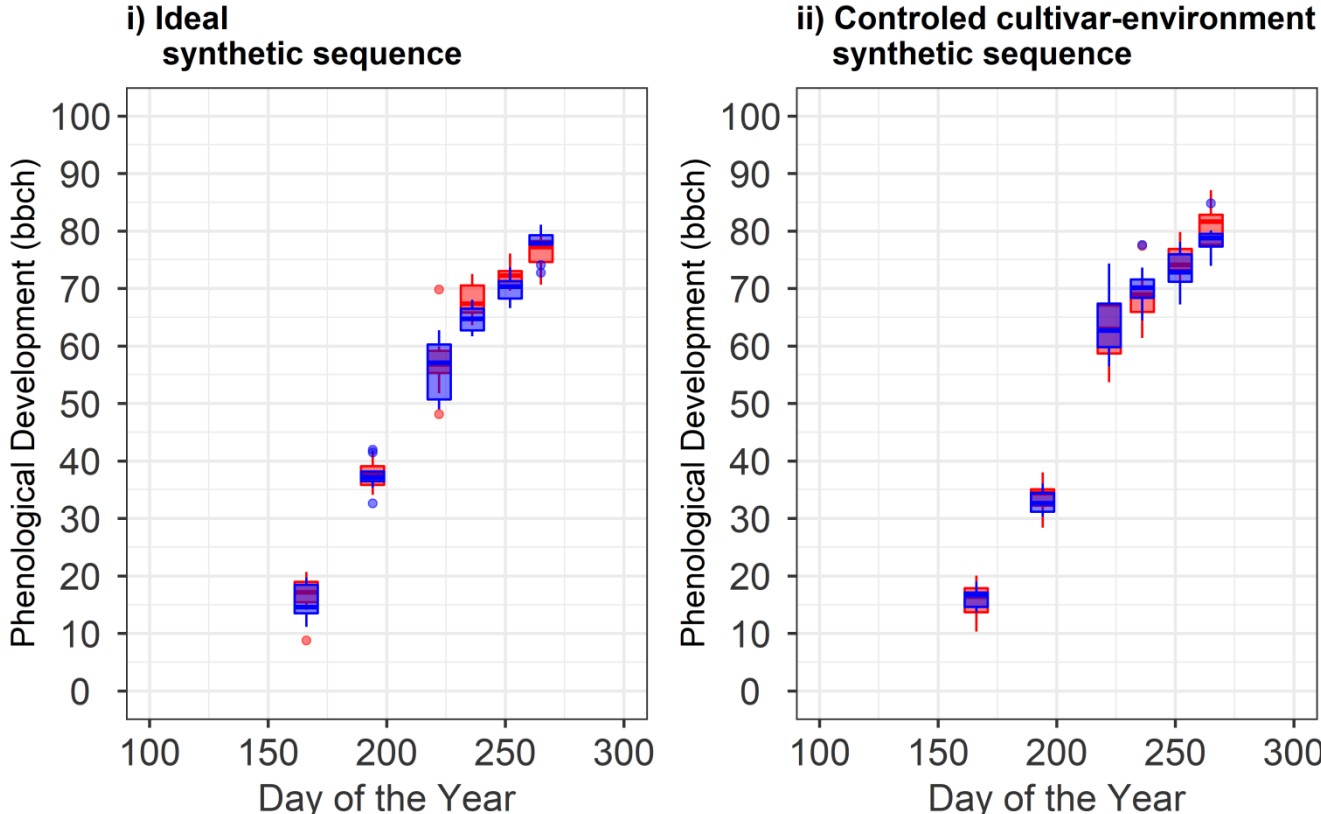


**Figure 3: Synthetic site-year observations used for calibration and prediction in (i) the ideal and (ii) controlled cultivar-environment synthetic sequences. The red box and whiskers represent the range of values for the 10 synthetic site-years used for calibration while the blue box and whiskers represent the range of values for the 10 site-years used for validation. Length of the box represents the inter-quartile range (IQR), whiskers extend from the box up to 1.5 × IQR and values beyond this range are**
**plotted as points.**

The *ideal* sequence represents a case in which the model is able to accurately simulate the observations. The only sources of difference between site-years are from the spatial variability at the field site which is below model resolution, and the incorrect identification of phenological stages during field observations. For generating the ideal sequence of site-years, we first calibrated the model to phenology at 6_2010. The parameter set $\theta_{MAP}$ corresponding to the maximum a posteriori

probability (MAP) estimate was used to simulate phenology and generate the synthetic dataset. To introduce inter-site-year differences, noise was added to simulated phenology $f(\theta_{MAP})_d$ at observation day d, where the noise was equal to the total observation uncertainty $\sigma_d$ at that day for site-year 6_2010. Thus, for each synthetic site-year on observation day d, the

phenological development was sampled from the range of total observation uncertainty $\sigma_d$ at 6_2010, around simulated

phenology $f(\theta_{MAP})_d$ . The synthetic observations were generated for the same observation days as the actual observations at

6_2010. We ensured that phenological development stages did not decrease with time, that is, $\dot{y}_d \geq \dot{y}_{d-1}$, where $\dot{y}_{d-1}$ is the

sampled phenological development at the previous observation day $d - 1$. Of the 20 site-years generated in this manner, 10

site-years were used for calibration while the remaining 10 were used for validation. The synthetic site-years were ordered

randomly during BSU calibration.

The *controlled cultivar-environment* sequence represents a sequence of site-years where the same cultivar is grown under the

same environmental conditions. In this case, however, the model may not accurately simulate the observations, implying the

presence of model structural error. For the controlled cultivar-environment sequence, we generated the synthetic site-year

data from observations of the cultivar grown at 6_2010. For each synthetic site-year, the phenological development $\dot{y}_d$ on

observation day d was sampled from the range of total observation uncertainty $\sigma_d$ around the observed mean $\overline{y}_d$. As in the

ideal sequence, we ensured that phenological development stages did not decrease with time. Again, 10 site-years were

randomly assigned for calibration.

### 2.6.2 True sequences

A total of 3 sequential updates in Kraichgau and 6 sequential updates in the Swabian Alb were analysed (Fig. 2). In each

sequential update, an additional site-year was included in the calibration dataset, following the actual chronological order in

which maize was grown in the regions. For the *Kraichgau sequence*, four site-years were available for calibration and

validation (3_2011, 2_2012 1_2014, and 2_2014). The model was sequentially calibrated to phenological development of

maize at site-years 3_2011, 2_2012 and 1_2014. After each update, phenological development was predicted for the

subsequent site-years. For example, in the first sequential update at Kraichgau, the model was calibrated to 3_2011. The site-

years 2_2012, 1_2014 and 2_2014 were used for validation to assess prediction quality of the calibrated model. In the second

sequential update, the model was calibrated to 3_2011 and 2_2012, while 1_2014 and 2_2014 were used for validation. Note

here that the number of site-years used for validation decrease with each sequential update. In the *Swabian Alb sequence*,

seven site-years were available for sequential calibration and validation (6_2010, 5_2011, 5_2012, 6_2013, 5_2015, 5_2016,

and 6_2016). The sequential updates were performed in a similar manner as in Kraichgau.



## 3 Results

In this section, we first describe the results for one example of Bayesian calibration using the data from site-year 6_2010

(section 3.1 Bayesian calibration results). Here we examine the resulting simulated phenology after calibration as well as the

posterior parameter distributions. We then look at the results from the synthetic and true sequences. We first evaluate the

evolution of the posterior parameter distributions with sequential updates. As an example, we analyse the marginal

distributions of the individual parameters and entropy of the joint parameter distributions for the true sequences (section 3.2

Parameter uncertainty). Lastly, we discuss the prediction quality results for the synthetic and true sequences (section 3.3

Prediction quality).

### 3.1 Bayesian calibration results

By way of example, Fig. 4 shows the Bayesian phenological model calibration results for silage maize at the first site-year

6_2010. Cross-plots of the posterior parameters (Fig. 4i) show weak negative correlation between PDD1 and TMINDEV1

and between PDD1 and DELTOPT1, while a weak positive correlation is observed between PDD1 and DELTMAX1. The

observed mean phenological development falls within the range of simulations after calibration (Fig. 4ii). The marginal

posterior parameter distributions are narrower than the initial prior distributions (Fig. 4iii). A shift in parameter distribution

to the margins of the prior ranges is also noted.





**Figure 4: Results of Bayesian calibration of the model to phenological development (BBCH stages) in site-year 6_2010. (i) Cross-**
**plot of the six posterior samples of the six estimated parameters. Red represents high density and blue low density. (ii) Observed**
**and simulated phenological development after calibration, plotted against the day of the year. The red points are the mean**
**observations, while the black error bars indicate +/- 3 standard deviations. The mean simulation is indicated by the continuous**
**black line. The green bands represent the different percentiles of simulated phenology. (iii) Prior (white) and posterior (salmon)**
**marginal parameter distributions for the six estimated parameters.**

## 3.2 Parameter uncertainty

We analysed the change in posterior parameter distribution with the sequential updates. Figure 5(i) shows the marginal initial

prior and posterior parameter distributions for the *Swabian Alb* and *Kraichgau true sequences*. The x-axis from left to right





indicates the initial prior parameter distribution followed by the sequential calibration of the model to an increasing number

of site-years. The distributions for the six estimated parameters are compared after each sequential update. The width of each

box with whiskers represents the uncertainty in the parameter values. There is a clear narrowing of parameter distributions

after the first sequential update from the initial prior. However, with the exception of DELTOPT2, the remaining parameters

do not show a noticeable and consistent narrowing in range with sequential updates. However, information entropy of the

joint posterior parameter distributions in Fig. 5(ii) decreases with sequential updates. There is a large reduction in entropy

after the first sequential update. In the Swabian Alb sequence (Fig. 5ii-a), entropy continues to reduce until the model is

calibrated to 6_2010, 5_2011 and 5_2012, after which there is no significant reduction. In the Kraichgau sequence (Fig. 5ii-

b), the inclusion of 1_2014 during calibration, results in further uncertainty reduction. Similar observations were made for

the synthetic sequences (supplementary material 5).



**(i) Marginal posterior parameter distributions**

(a) True sequence in Swabian Alb   (b) True sequence in Kraichgau





**Figure 5: (i) Marginal initial prior and posterior parameter distributions of the 6 estimated parameters plotted against the**
**calibration site-years, after BSU was applied to a true sequence (a) on the Swabian Alb and (b) in Kraichgau. The SPASS model was calibrated to observed phenological development (BBCH). (ii) Information entropy of the joint posterior parameter distributions plotted against the calibration site-years, after BSU was applied to the true sequences. The x-axis labels from left to right indicate the initial prior parameter distribution followed by the sequential calibration of the model to an increasing number of site-years. The '+' symbol before the site-year label on the x-axis indicates the new site-year that was included in the sequential**
**calibration. Length of the box in (i) represents the inter-quartile range (IQR), whiskers extend from the boxes up to 1.5 × IQR and values beyond this range are plotted as points.**

### 3.3 Prediction quality

### 3.3.1 Synthetic sequences

In the synthetic sequences, we assessed the prediction quality after applying BSU to 10 synthetic site-years, while excluding

model structural error and inter-site-year-differences in cultivar and environmental conditions in the *ideal* sequence and

*controlled cultivar-environment* sequence, respectively. In both sequences we account for identification uncertainty and

spatial variability within the modelled site. Figure 6 shows the trend in median NRMSE and bias with the sequential updates

from one to 10, for the two synthetic sequences. While the bias and NRMSE were calculated for all parameter vectors in the

posterior sample derived from the MCMC sampling method, only the median values are plotted and analysed for simplicity.

In the ideal sequence (Fig. 6i), the overall median NRMSE (Fig. 6i-a) and bias (Fig. 6i-b) are low, with many site-years

exhibiting a drop in the median NRMSE below a value of 1. However, after a few sequential updates no further reduction is

observed. In the controlled cultivar-environment sequence (Fig. 6ii), although most individual site-years showed a reduction

in median NRMSE with the sequential updates, there were some that exhibited an increase in median NRMSE (site-years 12

and 15 in Fig. 6ii-a). These site-years were also characterised by low initial median prediction bias, followed by an increase

in the absolute bias with sequential updates (Fig. 6ii-b).



Figure 6: (a) Median NRMSE and (b) median bias of prediction for the 10 validation site-years, after BSU was applied to the ideal (i) and controlled cultivar-environment (ii) synthetic sequences. The number of site-years used for calibration is shown on the x-axis and represents the sequential updates from one to 10. The SPASS model was calibrated to phenological development (BBCH). The lines and points correspond to the 10 synthetic validation site-years: ss1_11-ss1_20 from the ideal sequence and ss2_11-ss2_20 from the controlled cultivar-environment sequence.

### 3.3.2 True sequences

As a fewer number of site-years were used for validation in the true sequence as compared to the synthetic sequence, we analysed the prediction quality for each validation site-year individually, with the sequential updates. Figure 7 shows the prediction quality (i.e. NRMSE and bias for all the posterior predictive samples) of the model after BSU was applied to the true sequence of site-years in *Kraichgau* (Fig. 7i-iii) and on the *Swabian Alb* (Fig. 7iv-ix). For each site-year, we plot the

quality of prediction, after calibration to all preceding site-years. For example, Fig. 7(vi) shows the performance metric for site-year 6_2013 after the model was calibrated first to 6_2010, then to 6_2010 and 5_2011 and finally to 6_2010, 5_2011 and 5_2012, respectively (blue box-plots from left to right). As a reference, the performance metric derived from calibrating

the model to the target site-year, namely 6_2013 in Fig. 7(vi), is shown as the leftmost result (red box-plot) of each sequence. It is clear that this calibration always yields the best performance metrics for the given data. While the NBias was calculated for all parameter vectors in the posterior MCMC sample, only the median values of the absolute NBias are also plotted to compare the trends between NRMSE and NBias with the sequential updates.

The NRMSE is expected to decrease with the inclusion of more site-years for calibration. This holds true in the case of

Kraichgau, where mid-early cultivars were grown (Fig. 7ii, iii), but in hardly any case on Swabian Alb (Fig. 7iv-ix). We also expected the prediction quality to improve when a calibration sequence is made up of the same cultivar or ripening group. Note, however, the poor prediction quality in Fig. 7(iv) and the increase in NRMSE with the inclusion on 5_2011 in the calibration sequence in Fig. 7(ix). Additionally, the prediction quality for the early cultivar at 5_2016 (Fig. 7viii) deteriorates on the inclusion of the same cultivar grown at 5_2015 in the calibration sequence. In all predictions, the absolute NBias

follows a similar trend as the NRMSE. Note that there is a difference in the performance metrics between the different site-years when the model is directly calibrated to the target site-year (red box-plots in Fig. 7). The three site-years in Kraichgau and site-years 5_2011, 5_2012, 5_2015, and 6_2016 in Swabian Alb exhibit good to moderate calibration quality, while 6_2013 and 5_2016 have moderate to poor calibration quality.







**Figure 7: Performance metrics for site-years in Kraichgau (i-iii) and on Swabian Alb (iv-ix), after applying BSU to the two true sequences. The SPASS model was calibrated to observed phenological development (BBCH). NRMSE and bias are plotted against the site-years used in calibration. In each sub-plot, the red box-plot represents the calibration performance metric i.e. when the model is calibrated to site-year of interest. The blue box-plots represent the prediction performance metrics when the model is calibrated (from left to right) to an increasing number of preceding site-years. L, ME and E indicate the maturity group of the**
**cultivars: late, mid-early, and early, respectively. The '+' symbol before the site-year label on the x-axis and before the maturity group label indicates the new site-year that was included in the sequential calibration. Length of the box represents the inter-quartile range (IQR), whiskers extend from the box up to 1.5 × IQR and values beyond this range are plotted as points. The zero bias is indicated by a red dashed line in the bias plots. The median values of the absolute NBias are represented by red horizontal lines in the NRMSE plots.**

## 4 Discussion

In this study, we aimed to analyse whether progressively incorporating more data through Bayesian sequential updating (BSU) reduces model parameter uncertainty and produces robust parameter estimates for predicting phenology of silage maize.

### 4.1 Parameter uncertainty

Bayesian calibration resulted in reduced posterior parameter uncertainty in comparison to the initial prior ranges that were guided by expert knowledge (Fig. 4iii). The uncertainty in parameter DELTOPT2, reduced as seen from the narrowing of the marginal posterior distributions (Fig. 5). The remaining parameters did not show a consistent progressive reduction in uncertainty with the sequential updates. They also had a relatively higher correlation to the other parameters (Fig. 4i). The lack of uncertainty reduction may be due to equifinality, meaning that multiple parameter combinations produce the same

output (Adnan et al., 2020; He et al., 2017b; Lamsal et al., 2018). The reduction in information entropy of the posterior parameter distributions after the sequential updates (Fig.5ii) confirms the reduction in overall parameter uncertainty.

### 4.2 Prediction quality

We analysed synthetic sequences to assess whether a consistent reduction in prediction error is achieved when more site-years are available for calibration, in the absence of model structural errors (*ideal* sequence), and in the absence of inter-site-

year-differences due to cultivars and environmental conditions (*controlled cultivar-environment* sequence).

In the *ideal* sequence, the model was able to accurately simulate the observations, the only source of between-site-year-variability being within-site spatial variability and identification uncertainty. The overall initial prediction quality was moderate to good, indicating that when there was no model structural error, the calibrated model was able to predict

moderately well in spite of some observational variability (Fig. 6i). The progressive drop in median NRMSE to a value of 1

indicated that the calibrated model was able to explain all other variability apart from those arising from the total observation

uncertainty. Thus, with this sequence we demonstrated the successful application of BSU approach in ideal conditions.

In the *controlled cultivar-environment* sequence, the same cultivar was grown in the same environmental conditions across

the site-years. With this sequence, we tested the success of the BSU approach when model structural errors could exist in

addition to between-site-year-variability as in the ideal sequence. The overall change in prediction error reduced with the

sequential updates, as it possibly approaches an irreducible value. This is seen from the convergence of the different lines

corresponding to the prediction site-years in Fig. 6(ii-a). However, this irreducible value is higher than NRMSE of 1 due to

model structural error. Prediction error for most individual site-years reduced with the sequential updates. However, there

were two site-years where the error increased. These two site-years initially exhibited a low positive prediction bias that

progressively became negative with the sequential updates (Fig. 6ii-b). This can be attributed to representativeness of the

calibration data (Wallach et al., 2021). The two prediction site-years were more similar to the initial few site-years than the

later site-years in the calibration sequence.

We applied the BSU approach to real-world conditions represented by the true sequences of silage maize grown in

*Kraichgau* and on *Swabian Alb* (Fig. 7). In Kraichgau, the prediction quality improved with sequential updates as expected.

However, it deteriorated for many site-years on the Swabian Alb. This is again attributed to representativeness of the

calibration data as seen in the controlled cultivar-environment sequence. To understand this behaviour we carried out single

site-year calibration and predictions i.e. calibrating the model to individual site-years and predicting the remaining site-years

(for details refer to Appendix C: Single site-year calibration). As parameter estimates may vary by ripening group or cultivar,

we analysed the prediction results within these classes. Calibrating the model to a site-year from the same ripening group or

even the same cultivar as the prediction target site-year did not always result in the best prediction quality. Within the mid-

early and early ripening groups, prediction quality showed a correlation with the difference in average temperature during

the vegetative phase, between the calibration and prediction target site-year. This correlation indicated that the best

predictions of phenology for a particular site-year would be achieved when the model is calibrated to a cultivar from the

same ripening group and grown under the same temperature conditions during the vegetative phase. The calibration quality
for the individual site-years represented by red box-plots in Fig. 7, show that the model is able to simulate some site-years

better than others. Residual analysis (supplementary material 3) showed that the model was unable to capture the slow

development during the vegetative phase for these site-years with poorer calibration quality. This could be due to model

limitations (that is, model equations or hard-coded parameters) and could explain the correlation between temperature and

prediction quality.

The single site-year predictions showed that the mid-early cultivar grown at 1_2014 and 2_2014 were the best predictors of

each other and their prediction by the late cultivar at 3_2011 was poorer. Therefore, in case of the Kraichgau sequence (Fig.

7ii-iii) we observed a decrease in prediction error as we progressively calibrated the model to 3_2011, to 3_2011 and

2_2012, and to 3_2011, 2_2012 and 1_2014. In the Swabian Alb sequence (Fig. 7iv-ix) where mid-early and early cultivars

are grown, the effect of different ripening groups and temperatures caused an increase in prediction error.

In real-world conditions represented by the true sequences, the prediction quality thus depends on the interplay between

model limitations and inherent data structures presented in the differences between maturity group and cultivars. Since the

model calibration and prediction quality varies with environmental factors, it highlights the need to better account for the

influence of these environmental drivers in the model. This would increase model transferability to other sites. This could be

best achieved by improving the process representation in the model and by including the uncertainty in forcings during

calibration. An alternative approach would be to define separate cultivar- and environment-specific parameter distributions.

It is common practice to determine cultivar-specific parameters in crop modelling (Gao et al., 2020). He et al., (2017b) found

that data from different weather and site conditions are required to obtain a good calibrated parameter set for a particular

cultivar. Improved crop model performance has been reported upon the inclusion of environment-specific parameters in

calibration (Coelho et al., 2020). Cultivar- or genotype- and environment-specific parameters already exist in some models

(Jones et al., 2003; Wang et al., 2019). However, these genotype parameters have also been found to vary with the

environment, indicating that they may represent genotype × environment interactions and not fundamental genetic traits

(Lamsal et al., 2018). Further analysis of calibrated model parameters and model performance metrics with respect to

environmental variables would provide insights into areas for model improvement. Nonetheless, the cultivar and

environmental-dependency of parameters is a major drawback for large-scale model applications and long-term predictions,

as information on crop cultivars is usually not available on regional scales and specific characteristics of future cultivated

varieties are currently not known. While the collection of cultivar and maturity group information in official surveys is

essential, other Bayesian approaches, such as hierarchical Bayes, should be explored. Model calibration in a Bayesian

hierarchical framework would enable inherent data structures, represented by the cultivars within ripening groups of a

particular species, to be accounted for. Additionally, differences in environmental conditions can also be represented. On

regional scales, where information about maturity groups and cultivars are unavailable, accounting for environmental effects

alone may still prove to be beneficial. A Bayesian hierarchical approach could potentially be applied to predict the growth of

current as well as future cultivars.

### 4.3 Limitations

We would like to draw attention to the three assumptions in the current study which might cause an underestimation of

uncertainties. First, the standard deviation of the likelihood model was not estimated, but assumed to be known and equal to

sum of observed spatial variability and identification error. It represents the minimum error and is equal to the total error

only when there are no differences in environmental conditions and cultivars across the site-years. Second, the likelihood

model was assumed to be centred at zero, which only holds true when there are no structural errors. In most cases, however,

model structural errors and other systematic errors can exist, which could result in much larger errors than what was

assumed. Third, the errors are assumed to be independent and identically distributed. A violation of this assumption can lead

to underestimation of uncertainty in the parameters and the output state variable (Wallach et al., 2017). In the residual

analysis of the sequential updates with 3 or more site-years, a slight deviation from a Gaussian distribution was observed

(supplementary material 3). This skewness was caused due to model limitations, that is, its inability to capture the slow

development observed during the vegetative phase in some site-years. Autocorrelation of errors can exist for state variables

like phenology that are based on cumulative sums. However, based on the limited dataset an autocorrelation in the errors

could not be substantiated and an in-depth analysis is far beyond the scope of this study.

We observed that the posterior parameter distributions were at the margins of the initial prior distribution ranges, for which

this study now provides a basis to update this prior belief. This considerable update of the parameter prior indicates that

either the prior ranges are not suitable for the cultivars in this study, or that the parameters are compensating for structural

limitations of the model. Further in-depth investigation of their potential contributions could be achieved on much larger

datasets than the one at hand.

## 5 Conclusions

Through a Bayesian sequential updating (BSU) approach, we extended a classical application of Bayesian inference through time to analyse its effectiveness in the calibration and prediction of a crop phenology model. We assessed whether BSU of the SPASS model parameters, based on new observations made in different years, progressively improves prediction of

phenological development of silage maize.

We applied BSU to *synthetic* sequences and *true* sequences. As expected, the parameter uncertainty reduced in all sequences. The prediction errors reduced in most cases in the synthetic sequences, where we had an ideal model that was able to accurately simulate observations, and where the model could contain structural errors but the dataset contained only a single maize cultivar grown under the same environmental conditions. In the ideal synthetic sequence, the prediction quality was

variable for the first few sequential updates. The prediction error then reduced in both synthetic sequences until it approached an irreducible value. In the true sequences however, that included cultivars from different ripening groups and environmental conditions, the prediction quality deteriorated in most cases.  Differences in ripening group and temperature during the vegetative phase of growth between the calibration and prediction site-years influenced prediction quality.

With increasing amount of data being gathered and improvements in data-gathering techniques, there is a drive to use all

available data for model calibration. However, our study shows that a simplistic approach of updating the model parameter estimates without accounting for model limitations and inherent differences between datasets can lead to unsatisfactory predictions. To obtain robust parameter estimates for crop models applied on a large scale, the Bayesian approach needs to account for differences not only in maturity groups and cultivars but also environment. This could be achieved by applying Bayesian inference in a hierarchical framework, which will be a subject of future work.





**Appendix A: SPASS phenology model**

In the following paragraphs we describe the equations in the SPASS phenology model (Wang, 1997). The model parameters are indicated by words with all capitalized letters (e.g., SOWDEPTH, PDD1 etc.).

The crop passes through four main stages: sowing (stage -1.0), germination (stage -0.5), anthesis (stage 1.0, end of the vegetative phase and beginning of reproductive phase), and maturity (stage 2.0). Temperature and photoperiod are the two main factors affecting phenological development rate. The impact of water availability on germination is also reflected in the SPASS model.

For germination, soil moisture is the limiting factor. Germination occurs when:

$$\theta_{act(i_s)} > \theta_{pwp(i_s)} \qquad\qquad \text{A-1}$$

$$OR$$

$$0.02 \leq 0.65\left[\theta_{act(i_s)} - \theta_{pwp(i_s)}\right] + 0.35\left[\theta_{act(i_s+1)} - \theta_{pwp(i_s+1)}\right]$$

where $\theta_{act(i_s)}$ is the actual volumetric water content of the seed soil layer $i_s$ and $\theta_{pwp(i_s)}$ is the volumetric water content in the seed soil layer at permanent wilting point. If these conditions are not met within 40 days of sowing, crop failure is assumed.

The development rate from germination to emergence ($R_{dev,emerg}$) (d-1) is controlled by air temperature:

$$R_{dev,emerg} = (T_{avg} - T_{base}) \times 0.5/\Sigma T \qquad\qquad \text{A-2}$$

where, $T_{avg}$ (°C) is the daily average air temperature and $T_{base}$ (°C) is the base temperature set to 10°C for maize. The term $\Sigma T$ (°C) is the temperature sum needed for emergence:

$$\Sigma T = 15.0 + 6.0 \times SOWDEPTH \qquad\qquad \text{A-3}$$

where SOWDEPTH (cm) is the sowing depth of the seed.

After emergence, the development rate in the vegetative phase $R_{dev,v}$ (d-1) depends on temperature and photoperiod:

$$R_{dev,v} = R_{dev,v,max} \, f_{T,v} \, (T, TMINDEV1, TOPTDEV1, TMAXDEV1) \, f(h_{php}) \qquad \text{A-4}$$

where $R_{dev,v,max} = 1/PDD1$ is the maximum development rate in the vegetative phase (d-1), PDD1 is the number of physiological development days from emergence to anthesis (d), $f(h_{php})$ is the photoperiod factor, and



$f_{T,v}(T, TMINDEV1, TOPTDEV1, TMAXDEV1)$ is the temperature response function (TRF) for the vegetative phase. TMINDEV1, TOPTDEV1, and TMAXDEV1 are the minimum, optimum and maximum temperatures (°C) of the vegetative development phase, respectively. The photoperiod factor is expressed as:

$$f(h_{php}) = 1 - \exp\left(-4 \times (h_{php} - dlmin)/(DLOPT - dlmin)\right) \qquad \text{A-5}$$

where

$$dlmin = DLOPT + 4/PDL$$

$h_{php}$ (h) is the photoperiod length, that is, the amount of time between the beginning of the civil twilight before sunrise and the end of the civil twilight after sunset (the time when the true position of the centre of the sun is 4° below the horizon), PDL (-) is the photoperiod sensitivity and DLOPT (h) is the optimum daylength for a particular cultivar.

The development rate in the generative or reproductive phase ($R_{dev,r}$) (d-1) only depends on temperature such that:

$$R_{dev,r} = R_{dev,r,max} \, f_{T,r} \, (T, TMINDEV2, TOPTDEV2, TMAXDEV2) \qquad \text{A-6}$$

where $R_{dev,r,max} = 1/PDD2$ is the maximum development rate in the reproductive phase ( d-1), PDD2 is the number of physiological development days from anthesis to maturity (d) and $f_{T,r}(T, TMINDEV2, TOPTDEV2, TMAXDEV2)$ is the temperature response function (TRF) for the reproductive phase. TMINDEV2, TOPTDEV2, and TMAXDEV2 are the minimum, optimum and maximum temperatures (°C) of the reproductive development phase, respectively.

The temperature response function $f_T$ has cardinal temperatures: minimum temperature, TMINDEV (°C), optimum temperature, TOPTDEV (°C), and maximum temperature, TMAXDEV (°C):

$$f_T = \frac{2(T - TMINDEV)^\alpha (TOPTDEV - TMINDEV)^\alpha - (T - TMINDEV)^{2\alpha}}{(TOPTDEV - TMINDEV)^{2\alpha}} \qquad \text{A-7}$$

where

$$\alpha = \frac{\ln 2}{\ln\left[\dfrac{TMINDEV - TMAXDEV}{TOPTDEV - TMINDEV}\right]}$$

As the cardinal temperatures are phase-specific, the temperature response function is also phase-specific. For $f_{T,v}$, the cardinal temperatures are TMINDEV1, TOPTDEV1 and TMAXDEV1, while for $f_{T,r}$, the cardinal temperatures are TMINDEV2, TOPTDEV2 and TMAXDEV2.





The development stages after germination ($S_{dev}$) are calculated in daily time steps as:

$$S_{dev} = \sum_{d=d_{germ}}^{n} R_{dev} - 0.5 \qquad \text{A-8}$$

where $d_{germ}$ is the day on which seed germination occurs and n is the number of days after germination:

$$R_{dev} = \begin{cases} R_{dev,emerg} & \text{if} -0.5 \leq S_{dev} < 0.0 \\ R_{dev,v} & \text{if } 0.0 \leq S_{dev} < 1.0 \\ R_{dev,r} & \text{if } 1.0 \leq S_{dev} < 2.0 \end{cases} \qquad \text{A-9}$$

Finally, the SPASS development stages ($-0.5 \leq S_{dev} \leq 2$) are converted to BBCH development stages ($0 \leq \text{BBCH} \leq 95$).

Here, $S_{dev} = 0$ corresponds to BBCH $= 10$ (emergence and start of the vegetative phase), $S_{dev} = 0.4$ to BBCH $= 31$,

and $S_{dev} = 1$ to BBCH $= 61$ (start of the generative phase).

Preliminary simulations showed that the model was unable to capture the slow rate of emergence after sowing, as seen in the

observations, when the true sowing depth for maize was used. This could be due to uncertainty in the hard-coded parameters

in the emergence rate equations which were not estimated in this study. In order to simulate this slow emergence, an

effective sowing depth (SOWDEPTH) was set, which is deeper than the actual sowing depth range for maize (3-5cm).

In case of the cardinal temperatures for the vegetative and generative phases, the parameters $DELTOPT$ and

$DELTMAX$ were introduced instead of $TOPTDEV$ and $TMAXDEV$ during sensitivity analysis and MCMC sampling, to

ensure that during parameter sampling $TMINDEV < TOPTDEV < TMAXDEV$. Thus, $TMINDEV$, $DELTOPT$ and

$DELTMAX$ were used to parameterize the temperature response function during calibration, where $TOPTDEV =$

$TMINDEV + DELTOPT$ and $TMAXDEV = TOPTDEV + DELTMAX$.

**Appendix B: Posterior sampling using MCMC Metropolis algorithm**

The posterior parameter distribution was sampled using a Markov Chain Monte Carlo (MCMC) method based on the

Metropolis algorithm (Iizumi et al., 2009; Metropolis et al., 1953). Three Markov chains were run in parallel using the

foreach (Microsoft and Weston, 2020) and doParallel (Microsoft and Westen, 2019) packages in R (R Core Team, 2020).

First, initial parameter vectors were selected as a starting point for each chain. Then, the size of the transition kernel used to

propose new candidate parameter vectors in the chain was adapted, based on the acceptance rate, to improve the efficiency of



the MCMC algorithm (Gelman et al., 1996). After the adaptation, the Markov chains were run until the posterior parameter

distribution met the Gelman-Rubin convergence criteria (Brooks and Gelman, 1998; Gelman and Rubin, 1992). The detail

steps are given below:

**First sample:**

Step 1: Let $\theta_1$ be an arbitrary initial parameter vector in a chain, selected from within the parameter ranges provided by the

expert. This method of selection was used for the Bayesian calibration of site-year 6_2010. For the other calibration cases,

the initial parameter vectors were obtained by sampling from the range of the posterior parameter distribution after

calibration to 6_2010. This was done to reduce the time to convergence as it is expected that the posterior parameter

distributions for the other calibration cases would be in the vicinity of the posterior distribution obtained after calibration to

6_2010. The numerator of Bayes theorem is estimated as:

$$P(\theta_1|Y) \propto P(\theta_1)\,P(Y|\theta_1) \qquad\qquad \text{B-1}$$

where $P(Y|\theta_1)$  and $P(\theta_1)$  are calculated using the equations (8) and (9), respectively. The error function in Eq. (10)

required for $P(\theta_1)$ was calculated using the pracma package (Borchers, 2020).

**Jump Adaptation:**

A symmetrical transition kernel or jump distribution is used to select the next candidate parameter vector. The transition

kernel is a normal distribution that is centred at the current parameter vector, and has a variance vector $V^2$. The off-diagonal

elements of the variance-covariance matrix are zero.

Step 2: The transition kernel centred at $\theta_{t-1}$ is used to propose a new candidate parameter vector $\theta_t^*$.

Step 3: The model is simulated using parameter vector $\theta_t^*$ and the numerator of Bayes theorem is calculated using the prior

and likelihood as per equation B-1.

Step 4: The acceptance ratio (r) for a proposed candidate parameter vector is:

$$r = \frac{P(\theta_t^*)P(Y|\theta_t^*)}{P(\theta_{t-1})P(Y|\theta_{t-1})} \qquad\qquad \text{B-2}$$

Step 5: The candidate parameter vector $\theta_t^*$  is either accepted or rejected as the new parameter vector $\theta_t$ based on the

condition:



$$\theta_t = \begin{cases} \theta_t^* & r > u \\ \theta_{t-1} & r \leq u \end{cases} \qquad \qquad \text{B-3}$$

where $u \sim U(0,1)$ is a random sample from a uniform distribution between 0 and 1. Proposals of parameters which were outside the bounds of the prior or likelihood result in a zero in the numerator of equation B-2. These parameters are rejected and discarded. The next proposal is generated with the jump distribution centred at the last accepted parameter, until the next proposal is accepted.

Step 6: After 20 accepted parameter vectors per chain, the acceptance rate $ar = acc/tot$ is calculated across the chains, where $acc$ represents the number of accepted vectors (i.e. 20 accepted runs per chain × 3 chains in this case) and $tot$ represents the total vectors proposed. Based on the acceptance rate ($ar$), the standard deviation $V$ of the transition kernel, that controls the jump size, is adapted as per the condition in equation B-4, so that the acceptance rate is between 25% and 35% (Gelman et al., 1996; Tautenhahn et al., 2012).

$$V = \begin{cases} V \times 1.01 & ar \geq 0.35 \\ V \times 0.99 & ar \leq 0.25 \\ V & 0.25 < ar < 0.35 \end{cases} \qquad \text{B-4}$$

If the acceptance rate $ar$ is between 25% and 35%, we proceed to the main set of runs to obtain the posterior parameter distributions.

**Main runs:**

In the main runs, steps 2 to 5 are repeated with the final jump distribution achieved at the end of the jump adaptation steps.

Step 7: The convergence of the chains after jump adaptation, are checked using the Gelman-Rubin convergence criteria

(GR). The gelman.diag function from the coda package in R (Plummer et al., 2006) was used to evaluate the GR diagnostic after every 20 accepted parameter vectors in each chain. As per the GR diagnostic criteria the Markov chains have converged to represent a stable posterior distribution if within-chain variance is approximately equal to between-chain variance. The MCMC chains are stopped if there are a minimum of 500 accepted runs per chain and if GR <= 1.1 (Brooks and Gelman, 1998) for each parameter.

Step 8: In the final step, all the runs from the jump adaptation phase are discarded as burn-in. Parameters from the remaining accepted runs define the posterior distribution.

**Appendix C: Single site-year calibration**

In order to better understand the results of the true sequences, single site-year calibration and predictions were made within and across the two regions. As calibration yields the best performance metrics we analysed the median NRMSE ratio for

each prediction-target site-year, i.e., the ratio between the median NRMSE of prediction and the median NRMSE of calibration to the prediction-target (Fig. C-1). We expect that the model predicts best, i.e. with a low median NRMSE ratio, when it is calibrated to the same cultivar or ripening group. However, we found that this was not always the case. This is a result of careful analyses of calibration-prediction performance, detailed below.

The mid-early cultivar at 5_2011 was poorly predicted by all mid-early cultivars, but was better predicted by early cultivars.

The mid-early cultivar Grosso, grown at 1_2014 and 2_2014 in Kraichgau, were the best predictors of each other. However, even though the early cultivar, LG 30.217, was grown at 5_2015 and 5_2016 they were not the best predictors of each other. Similarly, the late cultivar Canavaro, grown in 2_2012 and 3_2011, were also not the best predictors of each other. In predictions for mid-early cultivars, a spread in median NRMSE ratio was seen when the model was calibrated to other mid-early cultivars. The mid-early cultivar at 1_2014 and 2_2014 in Kraichgau had a comparable prediction quality when the

model was calibrated to the late cultivar grown in Kraichgau or to the mid-early cultivars grown on the Swabian Alb.

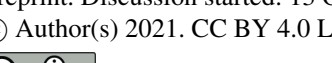



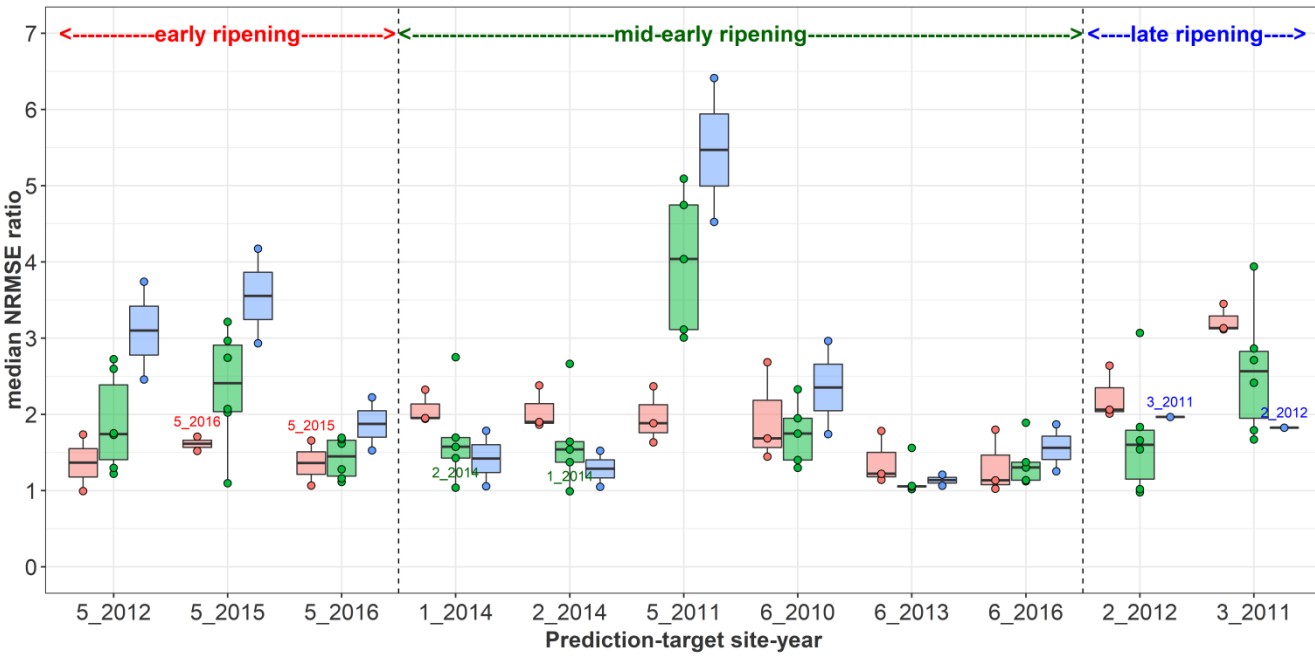

**Figure C-1:** Median NRMSE ratio for prediction-target site-years after single site-year calibration of the SPASS model to observed phenological development (BBCH). The median NRMSE ratio on the y-axis is the ratio between the median NRMSE of prediction and the median NRMSE of calibration to the prediction-target site-year. Each point represents the median NRMSE ratio of prediction of the site-year on the x-axis when the model was calibrated to phenology from every other site-year separately (single site-year calibration). The points are grouped and coloured by ripening group of the calibration site-year while the ripening group of the prediction target site-years are indicated on the top of the plot. The box and whiskers show the spread in median NRMSE ratio of predicting a particular site-year after the model was separately calibrated to site-years from a particular ripening group. Calibration site-year points from the same cultivar as the prediction site-year are labelled.

To explain the spread in prediction NRMSE within ripening groups, we examined the relationship between NRMSE and the difference in average temperature between the site-year used for calibration and the predicted or target site-year. The temperature was averaged over an interval of 40 to 100 days after sowing (i.e. approximate vegetative phase of development). For the mid-early ripening cultivars (Fig. C-2i), the median NRMSE shows a clear correlation. Albeit tested with a limited number of site-years, early-ripening cultivars (Fig. C-2ii) show a similar trend.





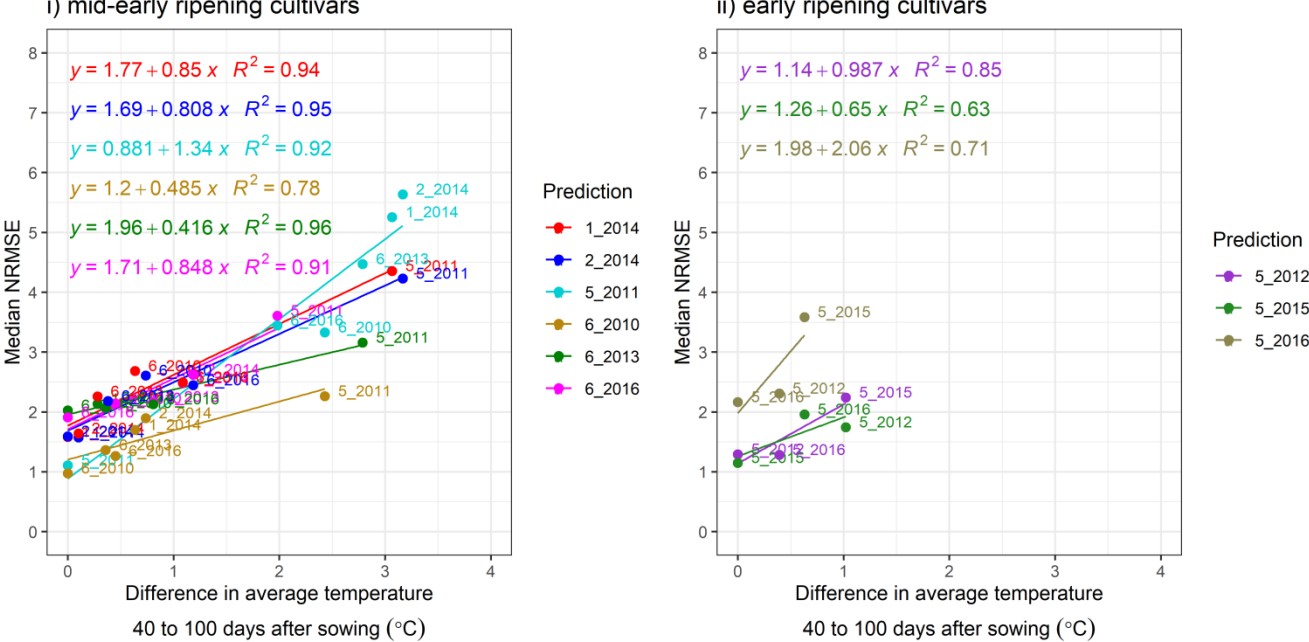


**Figure C-2:** **A cross-plot between the performance metric median NRMSE and the absolute difference in temperature between the site-year used for calibration and the prediction-target site-year, averaged over 40 to 100 days after sowing, for (i) mid-early and (ii) early ripening cultivars. Colours of the best-fit lines and points indicate the prediction-target site-year. Median NRMSE points at 0°C on the x-axis are calibration performance metrics for the target site-year while the remaining are prediction**
**performance metrics. Point labels indicate the site-years to which the model was calibrated. The SPASS model was calibrated to observed phenological development (BBCH).**

**6 Acknowledgements**

The contribution of Michelle Viswanathan was made possible through the Integrated Hydrosystem Modelling Research Training Group, funded by the German Research Foundation (DFG, GRK 1829). The contribution of Tobias K.D. Weber
was possible through the Collaborative Research Centre 1253 CAMPOS (Project 7: Stochastic Modelling Framework), funded by the German Research Foundation (DFG, Grant Agreement SFB 1253/1 2017). We would like to thank anonymous reviewers for suggestions and insightful comments that helped in improving the manuscript. We also thank Wolfgang Nowak and Joachim Ingwersen for their invaluable suggestions. The authors acknowledge support by the state of Baden-Wuerttemberg through the HPC cluster bwUniCluster (2.0).



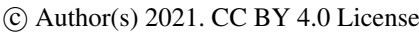



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
