# Peer review of "A Bayesian sequential updating approach to predict phenology of silage maize"

_Biogeosciences, 2021_

## Author Comment (AC1)

**Referee comments 1**

The authors present a very interesting approach to Bayesian model calibration that has been under-exploited within the crop modeling community. I very much enjoyed reading it. The topic and its treatment in this manuscript are compelling and likely of interest to the Biogeosciences readership and crop modeling community more generally. The results and discussion presented are interesting, but the sampling approach and checks for convergence were not documented well enough for me to evaluate whether the results presented were valid. Further documentation is needed before the manuscript can be reconsidered for publication.

We would like to thank you for your feedback and comments that have made us critically review our work and helped improve the manuscript. We are happy to hear that you enjoyed reading it. We address your comments in detail below. We have added more details regarding the sampling approach in the manuscript as well as in Supplementary Materials.

Our responses are in **bold**. Additions to the manuscript are marked as *MS*, *line...*: in *italics*. Additions to Supplementary Materials are marked as *Supp*: in *italics*.

Multiple details of the sampling approach used in this study remain unclear. The authors provide equations 3 & 4 as a formal expression of Bayesian sequential updating (BSU) in which the prior is defined based on a priori beliefs and the likelihood is derived from first site-year of data. Equation 4 indicates that the prior for the second site-year would then be the posterior distribution sampled using equation 3. The prior for the third site-year would be the posterior of the second site year, and so on. However, if I understand correctly, BSU is not the approach used in this study. Instead, the prior remains fixed across all site-year combinations and only the quantity of data used for the likelihood calculation increases with each subsequent site-year. This approach is broader and increasingly more likely to encompass the full range of environments over which prediction can be accurately performed. However, can this second approach be accurately termed BSU? I would suggest using an alternate term for this approach (at least something like "approximate BSU") and adjusting the title accordingly.

It is correct that in the actual implementation of BSU in this study, we keep the prior fixed across all the site-year combinations and only update the data in the likelihood estimation. Although this methodology is not strictly BSU, the results are a much better representation of the true approach than using posterior as the next prior. This is due to problems arising from approximating the posterior density in the strictly BSU approach. In fact, the results from using the strictly BSU approach would lead to larger approximations of the real results than the current approach. We therefore refrain from using the term 'approximate' in this case and are of the opinion that a change in terminology is unwarranted. We have, however, added the following sentence in the manuscript.

MS, line 197: We refer to the current methodology as BSU, although it is not strictly so, for reasons of simplicity and the formal similarity of our approach.

Still, that is a relatively minor point. The greater issue is the number of questions remaining on the how this general approach was implemented. For example:

We have added details of the approach in the manuscript and provided supporting plots in the Supplementary Materials that have also been included below. We have also addressed each question individually in the sections that follow.

We have added the following paragraph to the manuscript:

MS, line 231: The posterior parameter distribution was sampled using the Markov Chain Monte Carlo method – Metropolis algorithm (Metropolis et al., 1953) (for details refer to Appendix B: Posterior sampling using MCMC Metropolis algorithm). Three chains were run in parallel. A normal distribution was chosen as the transition kernel. The jump size was adapted so that the acceptance rate would be between 25% and 35% (A. Gelman et al., 1996; Tautenhahn et al., 2012). For each sequential update calibration case, when a new site-year was added to the calibration sequence, the three chains were re-initialized and the transition kernel was re-tuned. A preliminary calibration test case, in which the model was calibrated to site-year 6\_2010, was used to generate the starting points of the chains for each of the calibration cases. The starting points were randomly sampled from the posterior parameter range of the calibrated test case. This was done to reduce the time to convergence. For the test case calibration, the starting points of the chains were randomly sampled from the prior range. The number of iterations for adapting the transition kernel varied between the different calibration cases. This number was low for some of the calibration cases because we set the initial pre-adaptation value for the standard deviation of the transition kernel, so that the acceptance rate would be between 25% and 35%. This initial value was based on knowledge gained from preliminary calibration test simulations. Convergence of the chains after jump adaptation was checked using the Gelman-Rubin convergence diagnostic (Brooks & Gelman, 1998; Andrew Gelman & Rubin, 1992). The total number of samples of the posterior distribution in each calibration case was dependent on when the Gelman-Rubin diagnostic was <=1.1, while ensuring a minimum of 500 accepted samples per chain, that is, a minimum of 1500 samples across the three chains. In effect, the total number of samples per calibration case was greater than 1500. The burnin was variable and depended on the jump-adaptation. Only the iterations from the jump-adaptation step were discarded as burn-in. Parameter mixing was evaluated using trace-plots.

We have included the following details and table in Supplementary Materials S7.

Supp: The sequential update calibration cases for the true sequences in the Swabian Alb and in Kraichgau are listed in Table S7-1. The number of iterations required to adapt the jump-size (A) were variable (20-580) and dependent on the calibration case. In some cases this number was low because we set the initial pre-adaptation value for the standard deviation of the transition kernel so that the acceptance rate would be between 25% and 35%. This initial value was based on knowledge gained from preliminary calibration test runs. The jump adjustment factor (f) in Table S7-1 influences the standard deviation of the transition of the transition deviation of the prior parameter distributions taken from Table 2 in the main text. With N being the total number of iterations per chain, the total number of iterations across the three chains after burn-in is given by  $T = (N - A) \times 3$ .

On adding a new site-year, the chains were re-initialized and the transition kernel was re-tuned. New data was added to the dataset and the chains were allowed to adapt. The burn-in was variable and dependent of the jump-size adaptation. We ensured that a minimum of 500 accepted samples were generated per chain, that is, a minimum of 1500 total samples across chains were drawn. However, the actual number of samples drawn (T) was higher and dependent of when the Gelman-Rubin convergence diagnostic was <=1.1.

To assess parameter mixing, trace-plots were analysed (examples provided in Figure S27 and Figure S28). Additionally, auto-correlation plots (Figure S29, Figure S30) are provided (coda package in R (Plummer et al., 2006)) and effective sample size (ESS in Table S7-1) were calculated (mcmcse package in R (Flegal et al., 2021), (Vats et al., 2019)). Parameter DELTOPT2 generally showed good mixing and low auto-correlation. The effective sample size between 145 and 332, together with the Gelman-Rubin convergence diagnostic (<=1.1), provide sufficiently reliable posterior statistics for this study.

Table S7-1: MCMC sampling details for True sequence calibration cases in Kraichgau and the Swabian Alb

| Sequence                           | Calibration case | Number of
accepted runs
per chain
during jump
adaptation
(A) | Jump
adjustment
factor (f) | Total
accepted
samples
per chain
(N) | Total samples
after burn-in in
all chains $(T)$
= $(N - A) \times 3$ | ESS |
|------------------------------------|------------------|-----------------------------------------------------------------------------|----------------------------------|--------------------------------------------------|-------------------------------------------------------------------------------|-----|
| True
sequence
Swabian
Alb | 6_2010           | 20                                                                          | 3                                | 1480                                             | 4380                                                                          | 236 |

|                         | 6_2010, 5_2011                                       | 580 | 3.97  | 1100 | 1560  | 332 |
|-------------------------|------------------------------------------------------|-----|-------|------|-------|-----|
|                         | 6_2010, 5_2011,
5_2012                            | 20  | 5     | 800  | 2340  | 145 |
|                         | 6_2010, 5_2011,
5_2012, 6_2013                    | 40  | 4.95  | 820  | 2340  | 167 |
|                         | 6_2010, 5_2011,
5_2012, 6_2013,
5_2015         | 20  | 5     | 620  | 1800  | 196 |
|                         | 6_2010, 5_2011,
5_2012, 6_2013,
5_2015, 5_2016 | 240 | 6.7   | 1400 | 3480  | 159 |
| True Sequence Kraichgau | 3_2011                                               | 60  | 5.005 | 3280 | 9660  | 153 |
|                         | 3_2011, 2_2012                                       | 20  | 5     | 4480 | 13380 | 163 |
|                         | 3_2011, 2_2012,
1_2014                            | 20  | 7.7   | 5100 | 15240 | 168 |

- How were chains initialized? Randomly sampling the prior? (The effectiveness of the Gelman-Rubin diagnostic generally depends on the starting points for multiple chains be overdispersed with respect to the posterior.)
- MS, line 236: A preliminary calibration test case, in which the model was calibrated to site-year 6\_2010, was used to generate the starting points of the chains for each of the calibration cases. The starting points were randomly sampled from the posterior parameter range of the calibrated test case. This was done to reduce the time to convergence. For the test case calibration, the starting points of the chains were randomly sampled from the prior range.
- How many iterations were used for adapting the jump-size/transition kernel?
- MS, line 239: The number of iterations for adapting the transition kernel varied between the different calibration cases. This number was low for some of the calibration cases because we set the initial pre-adaptation value for the standard deviation of the transition kernel, so that the acceptance rate would be between 25% and 35%. This initial value was based on knowledge gained from preliminary calibration test simulations.
- When adding a new site-year, how were the chains handled? Were they re-initialized (along with retuning the transition kernel)? Was new data simply added to the dataset and chains allowed to adapt?

*MS, line 234:* For each sequential update calibration case, when a new site-year was added to the calibration sequence, the three chains were re-initialized and the transition kernel was re-tuned.

• How long was the warmup/burn-in? Was this variable?

*MS*, *line* 247: *The burn-in was variable and depended on the jump-adaptation. Only the iterations from the jump-adaptation step were discarded as burn-in.*

The iterations from the jump adaptation step varied by calibration case and are provided in Table S7-1 of the Supplementary Materials (see above) in column 'Number of accepted runs per chain during jump adaptation (A)'.

• How many samples were generated after warmup? I see a number of 500 in Appendix B. That seems very low.

The number of accepted samples (T) after warm-up varied by calibration case and are provided in Table S7-1 of the Supplementary Materials (see above) in column 'Total samples after burn-in in all chains  $(T) = (N - A) \times 3$ '.

- MS, line 244: The total number of samples of the posterior distribution in each calibration case was dependent on when the Gelman-Rubin diagnostic was

Figure S27: Trace-plots of 6 estimated parameters for the true sequence calibration of SPASS to phenology grown in the Swabian Alb at 6\_2010, 5\_2011, 5\_2012, 6\_2013, 5\_2015, and 5\_2016. The x-axis is the number of iterations and y-axis is the parameter. The colours indicate the three chains. The black solid vertical line indicates the burn-in phase during which the transition kernel was adapted.

---

## Author Comment (AC2)

**Referee comments 2**

The authors apply a Bayesian Sequential Updating approach to calibrate phenological parameters of a maize crop model. I haven't seen this type of approach applied to a crop model before and was very keen to learn more about it. In particular, given that long term and high quality agronomic data is often difficult to come by, the idea that the model could be iteratively updated as new data became available is very interesting. In general the quality of the writing is quite high. The manuscript was easy to read and mostly well explained. However, there for a few areas that I was unsure of and need to be further explained in order to make sure the methodology and results are valid.

**We would like to thank you for your feedback, insightful comments and for an in-depth review of our manuscript. Your comments were very valuable in clearly pin-pointing deficiencies and areas for improvement. We have addressed them in detail below and have also modified the manuscript and Supplementary Materials based on your feedback. We are delighted that you found the manuscript easy to read and understand. We hope that with these modifications and additional details, we were able to further improve it.**

**Our responses are in bold. Additions to the manuscript are marked as** *MS, line…:* **in** *italics***. Additions to Supplementary Materials are marked as** *Supp:* **in** *italics*.

I was confused as to why the authors chose to calibrate phenology parameters using data from different cultivars, which in several cases were known to have different phenologies. (i.e. early vs mid maturity). This isn't a standard practice unless perhaps you are trying to calibrate a model to match regional yields when you know a range of cultivars are used in the region. Or is there an assumption that Phenology doesn't differ between cultivars? I apologize if I've missed this, maize and this model are not my area of expertise. The authors do point out that this may have contributed to the decrease to model skill in the discussion but since it was known beforehand that there were different cultivars grown in different seasons I think the reasoning for this methological decision should be defined (e.g. what was the objective in calibrating phenology parameters if not to capture differences in phenology?)

**We have added the following paragraph in the Introduction to better highlight this study's application to regional modelling.**

*MS, line 84:* ***With this study, we explicitly deal with a well-known problem in regional modelling, which carries particular weight in the case of maize. On regional scale, maize cultivars may differ considerably in their phenological development, but cultivar information will rarely be available. Even if data on cultivars grown were available, phenological data on all relevant cultivars in a particular region will rarely be at hand. Consequently, model parameters are typically estimated for the crop species and not for the individual cultivars. Also, the maize cultivars of our study represent only a small subset of cultivars grown in Kraichgau and the Swabian Alb. We therefore grouped the maize cultivars into ripening groups for analysis of prediction quality.***

I was confused by the synthetic model runs and how the validation was assesed. There were to sythetic scenarios, an 'ideal' and a 'cultivar-environment' scenario. Both seem to be based on the 6_2010 site-year and simply to have had random noise added to them based on observed variance. So I'm not sure what the effective difference ended up being. Figure 3 seems to suggest that there isn't really any difference in the uncertainty between the two scenarios and there doesn't seem to be a difference between how they are simulated.

**The difference between ideal and controlled cultivar-environment sequences is in the ability of the model to accurately simulate the data. In the ideal sequence we use simulated phenology + noise to generate the synthetic dataset. In this case, there is no model structural error. In the controlled cultivar-environment scenario we use the observations + noise to generate the dataset. In this case, there is not only a random noise component but also a component of model structural error. As the noise and model error components cannot be resolved in this case, the model parameters compensate for both these components resulting in larger prediction errors.**

**We clarify this in the Discussion section.**

 *For the ideal sequence we used simulated phenology and added a random noise term that represents spatial variability and identification error. For the controlled cultivar-environment sequence we used the observations instead of simulated phenology to generate the dataset. Hence, in the latter sequence, there is not only random noise but also a model structural error component. As the noise and model error components cannot be resolved, the estimated model parameters compensate for both, leading to larger prediction errors (Fig. 6ii).*

Furthermore, the 'validation' set contains 10 'site-years' which are treated as independent (i.e. figure 6). From the methodology, these site-years are simply 10 random samples from the noise distribution. To me this seems that the mean and variance of these 10 would be more meaningful than treating them as individuals. Why are they treated separately?

**Although we agree with this point about the mean and variance, we treat them individually since the aim of the study is to mimic the yearly inflow of data and analyse how the posterior parameters distributions and prediction quality evolve for the next site-years.**

Specific Comments

Figure 1: What data are the boxplots based on? Is this 30 points per box (5 replicates by 6 locations) or 6 points per box (mean of 5 replicates for 6 locations)

**In figure 1(ii) we plot observations from one location i.e. site 6 in the year 2010, as an example. Each of the boxes and whiskers are based on 50 points corresponding to observations made on the same day i.e. 10 maize plants at 5 sub-plots within site 6 for one day in 2010. In site-year 6_2010, observations were made on 6 days during the growing season. So we have 6 boxes and whiskers corresponding to these 6 days. We now clarify this in the figure 1 caption.**

*Figure 1 caption: Each of the boxes and whiskers are based on 50 points corresponding to observations made on the same day i.e. 10 maize plants at 5 sub-plots within site 6 for one day in 2010. In site-year 6_2010, observations were made on 6 days during the growing season.*

Line 130: Probably need a bit of clarifying here. Was the simulation set to measured soil-water conditions at the start of each season? There is some mention of a 'burn-in' period presumably to settle the soil nutrient and water levels. My concern here was that some of the seasonal effects may come from what crops were grown beforehand and if there was any fertilization etc. that occurred at the start of the season.

**The simulation was set to measured soil-water conditions at the time of harvest of the previous crop. In most cases, maize was preceded by a cover crop and a fallow period. It was ensured that fertilizer application, tilling and other field management procedures that took place between the harvest of the previous crop and the sowing of maize were also incorporated in the model. However, it must be noted that fertilization is assumed to have no impact of maize phenology in the SPASS model used in the study. Soil water has an impact on phenology in the model, where emergence does not occur unless certain soil-water content conditions are met (Eq. A-1).**

Equation 12-13: Performance is based on skill throughout the season, for different site-years. Was there any analysis of skill at different times of the year across site-year combinations? I know this wouldn't make sense for the final sequence (only 1 validation site-year). But for the second site in particular where you could have run for 3 site-seasons and have 3-site-seasons for validation. It would be interesting to know if including the extra years had improved the ability across site-years (since this is what the calibration is actually trying to do). i.e. calibrate on site-year 1,2 and 3 tries to fit phenology to best explain variance across these three site-year combinations.

**The phenology measurements were taken at different times during the growing season and not on certain fixed dates across the site-years. Also, the same phenological development stages were not measured across the site-years. This made it difficult to analyse prediction bias and NRMSE within the growing season across the site-years with the sequential updates. However, we have now included a plot of the prediction residuals at the maximum a posteriori (MAP) parameter estimate for the true sequence in the Swabian Alb (Supplementary Materials S9). Here we see that the model does not predict the vegetative development well, in spite of including more site-years for calibration. The prediction quality changes are in line with the observations in Fig. 7 of the main text.**

**An excerpt from Supplementary Materials S9:**

*Supp: The prediction residuals are plotted against simulated phenology for the Swabian Alb true sequence at the maximum a posteriori probability (MAP) estimate of the model parameters (Figure S32). Here we analyse the prediction residuals within the growing season. The plots from left to right show the inclusion of the subsequent site-year for calibration. The model predicts poorly in the vegetative phase of development, in spite of including more site-years to the calibration sequence. The prediction residuals of the individual site-years are in agreement with the pattern observed in Figure 7 of the main text.*

[Figure]

*Figure S32: The plots from left to right show the inclusion of the subsequent site-year for calibration in the Swabian Alb sequence. Prediction residuals at the maximum a posteriori probability (MAP) estimate of the model parameters are plotted against simulated phenology. The points correspond to the prediction site-years. The zero residual reference is indicated by the red line.*

Figure 3: The boxplots should be side-by-side. It's difficult to read the overlapping boxes. In general across all figures, it's impossible to read the figures if they are not printed in colour. It would be worth checking with the journal if there are any requirements around this.

**We have now updated figure 3 and plotted the boxplots side-by-side. The updated plot is provided below.**

[Figure]

*Figure 3: Synthetic site-year observations used for calibration and prediction in (i) the ideal and (ii) controlled cultivar-environment synthetic sequences. The pink box and whiskers represent the range of values for the 10 synthetic site-years used for calibration while the blue box and whiskers represent the range of values for the 10 site-years used for validation. Length of the box represents the inter-quartile range (IQR), whiskers extend from the box up to 1.5 × IQR and values beyond this range are plotted as points.*

**Thank you for the feedback regarding the figures. We have now used labels and symbols in addition to the colours to make the figures grey-scale friendly. We provide the updated figures 6 and C-2 below as examples.**

[Figure]

*Figure 6: (a) Median NRMSE and (b) median bias of prediction for the 10 validation site-years, after BSU was applied to the ideal (i) and controlled cultivar-environment (ii) synthetic sequences. The number of site-years used for calibration is shown on the x-axis and represents the sequential updates from one to 10. The SPASS model was calibrated to phenological development (BBCH). The lines and points correspond to the 10 synthetic validation site-years: ss1_11-ss1_20 from the ideal sequence and ss2_11-ss2_20 from the controlled cultivar-environment sequence.*

[Figure]

*Figure C-2: A cross-plot between the performance metric median NRMSE and the absolute difference in temperature between the site-year used for calibration and the prediction-target site-year, averaged over 40 to 100 days after sowing, for (i) mid-early and (ii) early ripening cultivars. Colours of the best-fit lines and points indicate the prediction-target site-year. Median NRMSE points at 0°C on the x-axis are calibration performance metrics for the target site-year while the remaining are prediction performance metrics. Point labels indicate the site-years to which the model was calibrated. The SPASS model was calibrated to observed phenological development (BBCH).*

Line 285: 'structural model errors': What constitutes a structural model error? Maybe some examples for this particular model would be informative.

**We have now included an example of model structural error and a reference to the appendix where we provide more detail.**

*MS, line 309:… for example, the model's inability to capture slow emergence as explained in Appendix A: SPASS phenology model.*

Figure 4: I would like to see the PDFs for the sequential calibrations as a supplementary material. I think this would be more informative than say the entropy. For example, the first calibration is going to shift the distribution towards the specific cultivar. If the next calibration add a different cultivar (especially a different phenology group) the calibration has to try and shift somewhere between the two. This would either spread out the PDF (e.g. equi-finality as mentioned later by the authors) OR it may produce a bimodal distribution. (e.g. two separate but equally likely solutions that the calibration switches between while trying to match the two different cultivars). The results wouldn't change but it may help with interpretation.

**We opted to plot the entropy here instead of the marginal posterior pdfs as the entropy offers a better representation of the uncertainty reduction of the multivariate distribution with the sequential updates. With the marginal posterior pdfs, this is difficult to see due to parameter correlations. Yet, we have now included posterior pdfs for the true sequences in the Supplementary Material S5. We have also included a section in the discussion section where we interpret the results. We however express caution in interpreting the posterior parameters in terms of the cropping system due to parameter correlations and parameters possibly compensating for model structural errors.**

*MS, line 437: The optimum temperatures for vegetative (TOPTDEV1= TMINDEV1 + DELTOPT1) and reproductive (TOPTDEV2 = 8 + DELTOPT2) development are lower than our prior belief. The effective sowing depth (SOWDEPTH) is higher than the actual sowing depth of 3-5cm as the model cannot capture slow emergence (as discussed in the Appendix A: SPASS phenology model). In Kraichgau, the posterior distributions for SOWDEPTH and minimum temperature for vegetative development (TMINDEV1) did not change significantly as compared to the prior, indicating that*

*the model did not learn much from the data. These parameters, however, show a change from the prior in the Swabian Alb. Kraichgau is warmer than the Swabian Alb. On most days, temperatures in Kraichgau are above the minimum temperature for vegetative development (TMINDEV1), resulting in limited learning. A similar reasoning applies to SOWDEPTH which is a proxy parameter that impacts emergence rate. Emergence occurs only above a certain threshold temperature which is hard-coded in the model. Temperatures in Kraichgau are mostly above this threshold temperature for emergence, resulting in limited learning and insignificant change from the prior distribution. In the Kraichgau sequence (Fig. 5i-b), PDD1 and DELTMAX1 decrease when site-year 1_2014 is added to the calibration sequence. Both parameters cause a faster development rate during the vegetative phase. This faster vegetative development results in earlier initiation of the reproductive phase, as seen in the mid-early ripening cultivar 1_2014 as compared to the late cultivars 3_2011 and 2_2012. In the Swabian Alb sequence (Fig. 5i-a), inclusion of early cultivars at 5_2012 and 5_2016 results in shallower SOWDEPTH and consequently, faster emergence. However, whether this early emergence is truly a feature of early cultivars or a consequence of the timing of first observations in the growing season cannot be satisfactorily distinguished with the available data. The physiological development days at optimum vegetative phase temperature (PDD1) were also lower than our initial prior belief. We, however, interpret these results with caution as parameters may compensate for model structural errors and some parameters are correlated (Alderman & Stanfill, 2017).*

**We provide an excerpt below from Supplementary Materials S5 with the posterior pdfs for the true sequences.**

*Supp: Marginal posterior probability density functions (pdf) of the 6 estimated parameters for the true sequences in Kraichgau (Figure S23) and the Swabian Alb (Figure S24) are provided. Refer to the main text (Discussion: Parameter Uncertainty) for details.*

[Figure]

*Figure S23: Marginal posterior probability density functions of the 6 estimated parameters after BSU in the true Kraichgau sequence. The y-axis read from bottom to top represent the site-year that was added in the sequential update, corresponding to each density plot. The parameter values are on the x-axis.*

[Figure]

*Figure S24: Marginal posterior probability density functions of the 6 estimated parameters after BSU in the true Swabian Alb sequence. The y-axis read from bottom to top represent the site-year that was added in the sequential update, corresponding to each density plot. The parameter values are on the x-axis.*

Figure 6: as mentioned above, if the 10 validation sites are just random permutations I don't think showing each of the ten is meaningful. The average and perhaps the variance would suffice.

**We plot the prediction quality of the individual site-years in the synthetic sequences because we compare their behaviour to the prediction quality of individual site-years in the true sequences. We specifically highlight the trend of site-years ss2_12 and ss2_15 in the controlled cultivar-environment sequence which is attributed to the representativeness of the calibration data. Poor representativeness (due to different ripening groups/environments) of the calibration data is also observed in the true sequence, resulting in unreliable predictions.**

Figure 7: This figure was difficult to read. As the validation results are what is important, I don't think the calibration (red box) is necessary. I think the meaning of the boxes needs to be more clearly explained as well. The boxes represent different MCMC initializations? Line 376 suggests that only the median value is plotted.

**The red box-plots serve as a reference of how well the model can perform with the given data. We have now made them less conspicuous in the figure, so as to avoid distracting the reader. The median values of absolute Nbias (red asterisk) and the red dashed line indicating zero bias, as described in the text and caption, were missing from the plots and may have caused the confusion. We apologise for the oversight and have now corrected this plot. The updated plot is provided below.**

[Figure]

*Figure 7: Performance metrics for site-years in Kraichgau (i-iii) and on Swabian Alb (iv-ix), after applying BSU to the two true sequences. The SPASS model was calibrated to observed phenological development (BBCH). NRMSE and bias are plotted against the site-years used in calibration. In each sub-plot, the grey box-plot represents the calibration performance metric i.e. when the model is calibrated to site-year of interest. The blue box-plots represent the prediction performance metrics when the model is calibrated (from left to right) to an increasing number of preceding site-years. L, ME and E indicate the maturity group of the cultivars: late, mid-early, and early, respectively. The '+' symbol before the site-year label on the x-axis and*

*before the maturity group label indicates the new site-year that was included in the sequential calibration. Length of the box represents the inter-quartile range (IQR), whiskers extend from the box up to 1.5 × IQR and values beyond this range are plotted as points. The zero bias is indicated by a red dashed line in the bias plots. The median values of the absolute NBias are represented by red asterisks (*) in the NRMSE plots.*

The specifications of the MCMC process aren't clearly defined in the Methodology. (e.g. number of initializations, burn-in period etc.).

**We have now provided specifications of the MCMC process in the Methodology section.**

*MS, line 231: The posterior parameter distribution was sampled using the Markov Chain Monte Carlo method – Metropolis algorithm (Metropolis et al., 1953) (for details refer to Appendix B: Posterior sampling using MCMC Metropolis algorithm). Three chains were run in parallel. A normal distribution was chosen as the transition kernel. The jump size was adapted so that the acceptance rate would be between 25% and 35% (A. Gelman et al., 1996; Tautenhahn et al., 2012). For each sequential update calibration case, when a new site-year was added to the calibration sequence, the three chains were re-initialized and the transition kernel was re-tuned. A preliminary calibration test case, in which the model was calibrated to site-year 6_2010, was used to generate the starting points of the chains for each of the calibration cases. The starting points were randomly sampled from the posterior parameter range of the calibrated test case. This was done to reduce the time to convergence. For the test case calibration, the starting points of the chains were randomly sampled from the prior range. The number of iterations for adapting the transition kernel varied between the different calibration cases. This number was low for some of the calibration cases because we set the initial pre-adaptation value for the standard deviation of the transition kernel, so that the acceptance rate would be between 25% and 35%. This initial value was based on knowledge gained from preliminary calibration test simulations. Convergence of the chains after jump adaptation was checked using the Gelman-Rubin convergence diagnostic (Brooks & Gelman, 1998; Andrew Gelman & Rubin, 1992). The total number of samples of the posterior distribution in each calibration case was dependent on when the Gelman-Rubin diagnostic was <=1.1, while ensuring a minimum of 500 accepted samples per chain, that is, a minimum of 1500 samples across the three chains. In effect, the total number of samples per calibration case was greater than 1500. The burn-in was variable and depended on the jump-adaptation. Only the iterations from the jump-adaptation step were discarded as burn-in. Parameter mixing was evaluated using trace-plots.*

**We have also provided more details like the number of jump-adaptation iterations, burn-in, total samples, effective sample size, trace-plots, and auto-correlation plots in the Supplementary Material S7.**

*Supp: The sequential update calibration cases for the true sequences in the Swabian Alb and in Kraichgau are listed in Table S7-1. The number of iterations required to adapt the jump-size (A) were variable (20-580) and dependent on the calibration case. In some cases this number was low because we set the initial pre-adaptation value for the standard deviation of the transition kernel so that the acceptance rate would be between 25% and 35%. This initial value was based on knowledge gained from preliminary calibration test runs. The jump adjustment factor ($f$) in Table S7-1 influences the standard deviation of the transition kernel (V) such that $V = \frac{sd}{f}$ where $sd$ is the standard deviation of the prior parameter distributions taken from Table 2 in the main text. With N being the total number of iterations per chain, the total number of iterations across the three chains after burn-in is given by $T = (N - A) \times 3$.*

*On adding a new site-year, the chains were re-initialized and the transition kernel was re-tuned. New data was added to the dataset and the chains were allowed to adapt. The burn-in was variable and dependent of the jump-size adaptation. We ensured that a minimum of 500 accepted samples were generated per chain, that is, a minimum of 1500 total samples across chains were drawn. However, the actual number of samples drawn (T) was higher and dependent of when the Gelman-Rubin convergence diagnostic was <=1.1.*

*To assess parameter mixing, trace-plots were analysed (examples provided in Figure S27 and Figure S28). Additionally, auto-correlation plots (Figure S29, Figure S30) are provided (coda package in R (Plummer et al., 2006)) and effective sample size (ESS in Table S7-1) were calculated (mcmcse package in R (Flegal et al., 2021), (Vats et al., 2019)). Parameter DELTOPT2 generally showed good mixing and low auto-correlation. The effective sample size between 145 and 332,*

*together with the Gelman-Rubin convergence diagnostic (<=1.1), provide sufficiently reliable posterior statistics for this study.*

*Table S7-1: MCMC sampling details for True sequence calibration cases in Kraichgau and the Swabian Alb*

| Sequence | Calibration case | Number of accepted runs per chain during jump adaptation (A) | Jump adjustment factor (f) | Total accepted samples per chain (N) | Total samples after burn-in in all chains (T) $= (N - A) \times 3$ | ESS |
|---|---|---|---|---|---|---|
| *True sequence Swabian Alb* | 6_2010 | 20 | 3 | 1480 | 4380 | 236 |
| | 6_2010, 5_2011 | 580 | 3.97 | 1100 | 1560 | 332 |
| | 6_2010, 5_2011, 5_2012 | 20 | 5 | 800 | 2340 | 145 |
| | 6_2010, 5_2011, 5_2012, 6_2013 | 40 | 4.95 | 820 | 2340 | 167 |
| | 6_2010, 5_2011, 5_2012, 6_2013, 5_2015 | 20 | 5 | 620 | 1800 | 196 |
| | 6_2010, 5_2011, 5_2012, 6_2013, 5_2015, 5_2016 | 240 | 6.7 | 1400 | 3480 | 159 |
| *True Sequence Kraichgau* | 3_2011 | 60 | 5.005 | 3280 | 9660 | 153 |
| | 3_2011, 2_2012 | 20 | 5 | 4480 | 13380 | 163 |
| | 3_2011, 2_2012, 1_2014 | 20 | 7.7 | 5100 | 15240 | 168 |

[Figure]

*Figure S27: Trace-plots of 6 estimated parameters for the true sequence calibration of SPASS to phenology grown in the Swabian Alb at 6_2010, 5_2011, 5_2012, 6_2013, 5_2015, and 5_2016. The x-axis is the number of iterations and y-axis is the parameter. The colours indicate the three chains. The black solid vertical line indicates the burn-in phase during which the transition kernel was adapted.*

[Figure]

*Figure S28: Trace-plots of 6 estimated parameters for the true sequence calibration of SPASS to phenology grown in Kraichgau at 3_2011 and 2_2012. The x-axis is the number of iterations and y-axis is the parameter. The colours indicate the three chains. The black solid vertical line indicates the burn-in phase during which the transition kernel was adapted.*

[Figure]

*Figure S29: Auto-correlation plots of 6 estimated parameters for the true sequence calibration of SPASS to phenology grown in the Swabian Alb at 6_2010, 5_2011, 5_2012, 6_2013, 5_2015, and 5_2016. The x-axis is the lag distance and y-axis is the auto-correlation. The colours indicate the three chains.*

[Figure]

*Figure S30: Auto-correlation plots of 6 estimated parameters for the true sequence calibration of SPASS to phenology grown in Kraichgau at 3_2011 and 2_2012. The x-axis is the lag distance and y-axis is the auto-correlation. The colours indicate the three chains.*

Line 460: "it is common practice to determine cultivar-specific parameters in crop modelling", Because of this I think some explanation is needed as to why phenology was calibrated across cultivars.

**We have now clarified in the Introduction that this study applies to regional modelling where multiple cultivars are grown.**

*MS, line 84:* *With this study, we explicitly deal with a well-known problem in regional modelling, which carries particular weight in the case of maize. On regional scale, maize cultivars may differ considerably in their phenological development, but cultivar information will rarely be available. Even if data on cultivars grown were available, phenological data on all relevant cultivars in a particular region will rarely be at hand. Consequently, model parameters are typically estimated for the crop species and not for the individual cultivars. Also, the maize cultivars of our study represent only a small subset of cultivars grown in Kraichgau and the Swabian Alb. We therefore grouped the maize cultivars into ripening groups for analysis of prediction quality.*

Line 470: "While the collection of cultivar and maturity group information…." I'm not sure of the link here to the hierarchical Bayes method? Are you trying to say that ideally cultivar values should be collected but an alternative is to calibrate using Hierarchical Bayes? Why not do that using your technique?

**We have now rephrased this statement.**

*MS, line 520:* *Collection of cultivar and maturity group information in official surveys is essential. Furthermore, other Bayesian approaches such as hierarchical Bayes, which allow for the incorporation of this information during calibration, should be explored.*

**References**

Alderman, P. D., & Stanfill, B. (2017). Quantifying model-structure- and parameter-driven uncertainties in spring wheat phenology prediction with Bayesian analysis. *European Journal of Agronomy*, *88*, 1–9. https://doi.org/10.1016/j.eja.2016.09.016

Brooks, S. P., & Gelman, A. (1998). General Methods for Monitoring Convergence of Iterative Simulations. *Journal of Computational and Graphical Statistics*, *7*(4), 434–455. https://doi.org/10.1080/10618600.1998.10474787

Flegal, J. M., Hughes, J., Vats, D., Dai, N., Gupta, K., & Maji, U. (2021). mcmcse: Monte Carlo Standard Errors for MCMC. Riverside, CA, and Kanpur, India.

Gelman, A., Roberts, G. O., & Gilks, R. W. (1996). Efficient Metropolis jumping rules. In J. M. Bernardo, J. O. Berger, A. P. Dawid, & A. F. M. Smith (Eds.), *Bayesian Statistics* (Vol. 5, pp. 599–608). Oxford University Press.

Gelman, Andrew, & Rubin, D. (1992). Inference from iterative simulation using multiple sequences. *Statistical Science*, *7*(4), 457–511. Retrieved from https://projecteuclid.org/euclid.ss/1177011136

Metropolis, N., Rosenbluth, A. ., Rosenbluth, M. ., & Teller, A. . (1953). Equation of State Calculations by Fast Computing Machines. *The Journal of Chemical Physics*, *21*(6). Retrieved from https://bayes.wustl.edu/Manual/EquationOfState.pdf

Plummer, M., Best, N., Cowles, K., & Vines, K. (2006). CODA: Convergence Diagnosis and Output Analysis for MCMC. *R News*, *6*(1), 7–11. Retrieved from https://journal.r-project.org/archive/

Tautenhahn, S., Heilmeier, H., Jung, M., Kahl, A., Kattge, J., Moffat, A., & Wirth, C. (2012). Beyond distance-invariant survival in inverse recruitment modeling: A case study in Siberian Pinus sylvestris forests. *Ecological Modelling*, *233*, 90–103. https://doi.org/10.1016/j.ecolmodel.2012.03.009

Vats, D., Flegal, J. M., & Jones, G. L. (2019). Multivariate output analysis for Markov chain Monte Carlo. *Biometrika*, *106*(2), 321–337. https://doi.org/10.1093/biomet/asz002

**S9. Prediction residuals within season**

The prediction residuals are plotted against simulated phenology for the Swabian Alb true sequence at the maximum a posteriori probability (MAP) estimate of the model parameters (Figure S32). Here we analyse the prediction residuals within the growing season. The plots from left to right show the inclusion of the subsequent site-year for calibration. The model predicts poorly in the vegetative phase of development, in spite of including more site-years to the calibration sequence. The prediction residuals of the individual site-years are in agreement with the pattern observed in Figure 7 of the main text.

[Figure]

Figure S32     The plots from left to right show the inclusion of the subsequent site-year for calibration in the Swabian Alb sequence. Prediction residuals at the maximum a posteriori probability (MAP) estimate of the model parameters are plotted against simulated phenology. The points correspond to the prediction site-years. The zero residual reference is indicated by the red line.

**S5. Marginal posterior parameter pdf: true sequences**

Marginal posterior probability density functions (pdf) of the 6 estimated parameters for the true sequences in Kraichgau (Figure S23) and the Swabian Alb (Figure S24) are provided. Refer to the main text (Discussion: Parameter Uncertainty) for details.

[Figure]

Figure S23     Marginal posterior probability density functions of the 6 estimated parameters after BSU in the true Kraichgau sequence. The y-axis read from bottom to top represent the site-year that was added in the sequential update, corresponding to each density plot. The parameter values are on the x-axis.

[Figure]

Figure S24      Marginal posterior probability density functions of the 6 estimated parameters after BSU in the true Swabian Alb sequence. The y-axis read from bottom to top represent the site-year that was added in the sequential update, corresponding to each density plot. The parameter values are on the x-axis.

**S7. MCMC diagnostics**

The sequential update calibration cases for the true sequences in the Swabian Alb and in Kraichgau are listed in Table S7-1. The number of iterations required to adapt the jump-size (A) were variable (20-580) and dependent on the calibration case. In some cases this number was low because we set the initial pre-adaptation value for the standard deviation of the transition kernel so that the acceptance rate would be between 25% and 35%. This initial value was based on knowledge gained from preliminary calibration test runs. The jump adjustment factor (f) in Table S7-1 influences the standard deviation of the transition kernel (V) such that $V = \frac{sd}{f}$ where sd is the standard deviation of the prior parameter distributions taken from Table 2 in the main text. With N being the total number of iterations per chain, the total number of iterations across the three chains after burn-in is given by $T = (N - A) \times 3$.

On adding a new site-year, the chains were re-initialized and the transition kernel was re-tuned. New data was added to the dataset and the chains were allowed to adapt. The burn-in was variable and dependent of the jump-size adaptation. We ensured that a minimum of 500 accepted samples were generated per chain, that is, a minimum of 1500 total samples across chains were drawn. However, the actual number of samples drawn (T) was higher and dependent of when the Gelman-Rubin convergence diagnostic was <=1.1.

To assess parameter mixing, trace-plots were analysed (examples provided in Figure S27 and Figure S28). Additionally, auto-correlation plots (Figure S29, Figure S30) are provided (coda package in R (Plummer et al., 2006)) and effective sample size (ESS in Table S7-1) were calculated (mcmcse package in R (Flegal et al., 2021), (Vats et al., 2019)). Parameter DELTOPT2 generally showed good mixing and low auto-correlation. The effective sample size between 145 and 332, together with the Gelman-Rubin convergence diagnostic (<=1.1), provide sufficiently reliable posterior statistics for this study.

Table S7-1: MCMC sampling details for True sequence calibration cases in Kraichgau and the Swabian Alb

| Sequence | Calibration case | Number of accepted runs per chain during jump adaptation (A) | Jump adjustment factor (f) | Total accepted samples per chain (N) | Total samples after burn-in in all chains (T) = $(N - A) \times 3$ | ESS |
|---|---|---|---|---|---|---|
| True sequence Swabian Alb | 6_2010 | 20 | 3 | 1480 | 4380 | 236 |
| | 6_2010, 5_2011 | 580 | 3.97 | 1100 | 1560 | 332 |
| | 6_2010, 5_2011, 5_2012 | 20 | 5 | 800 | 2340 | 145 |
| | 6_2010, 5_2011, 5_2012, 6_2013 | 40 | 4.95 | 820 | 2340 | 167 |
| | 6_2010, 5_2011, 5_2012, 6_2013, 5_2015 | 20 | 5 | 620 | 1800 | 196 |
| | 6_2010, 5_2011, 5_2012, 6_2013, 5_2015, 5_2016 | 240 | 6.7 | 1400 | 3480 | 159 |
| True Sequence Kraichgau | 3_2011 | 60 | 5.005 | 3280 | 9660 | 153 |
| | 3_2011, 2_2012 | 20 | 5 | 4480 | 13380 | 163 |
| | 3_2011, 2_2012, 1_2014 | 20 | 7.7 | 5100 | 15240 | 168 |

[Figure]

Figure S27    Trace-plots of 6 estimated parameters for the true sequence calibration of SPASS to phenology grown in the Swabian Alb at 6_2010, 5_2011, 5_2012, 6_2013, 5_2015, and 5_2016. The x-axis is the number of iterations and y-axis is the parameter. The colours indicate the three chains. The black solid vertical line indicates the burn-in phase during which the transition kernel was adapted.

[Figure]

Figure S28        Trace-plots of 6 estimated parameters for the true sequence calibration of SPASS to phenology grown in Kraichgau at 3_2011 and 2_2012. The x-axis is the number of iterations and y-axis is the parameter. The colours indicate the three chains. The black solid vertical line indicates the burn-in phase during which the transition kernel was adapted.

[Figure]

Figure S29    Auto-correlation plots of 6 estimated parameters for the true sequence calibration of SPASS to phenology grown in the Swabian Alb at 6_2010, 5_2011, 5_2012, 6_2013, 5_2015, and 5_2016. The x-axis is the lag distance and y-axis is the auto-correlation. The colours indicate the three chains.

[Figure]

Figure S30  Auto-correlation plots of 6 estimated parameters for the true sequence calibration of SPASS to phenology grown in Kraichgau at 3_2011 and 2_2012. The x-axis is the lag distance and y-axis is the auto-correlation. The colours indicate the three chains.